



Probing key organic substances driving new particle growth initiated by
iodine nucleation in coastal atmosphere
Yibei Wan,[1] Xiangpeng Huang,[2] Bin Jiang,[3] Binyu Kuang,[4] Manfei Lin,[4] Deming
Xia[5], Yuhong Liao,[3] Jingwen Chen,[5] Jianzhen Yu,[4] and Huan Yu[1]
[1] Department of Atmospheric Science, School of Environmental Studies, China
University of Geosciences, Wuhan, 430074, China
[2] School of Environmental Science and Engineering, Nanjing University of Information
Science and Technology, Nanjing, 210044, China
[3] Guangzhou Institute of Geochemistry, Chinese Academy of Sciences, Guangzhou
510640, China
[4] Department of Chemistry, Hong Kong University of Science & Technology, Clear
Water Bay, Kowloon, Hong Kong, China
[5] School of Environmental Science and Technology, Dalian University of Technology,
Dalian 116024, China
Corresponding author: H. Yu (yuhuan@cug.edu.cn)



**Graphic abstract**

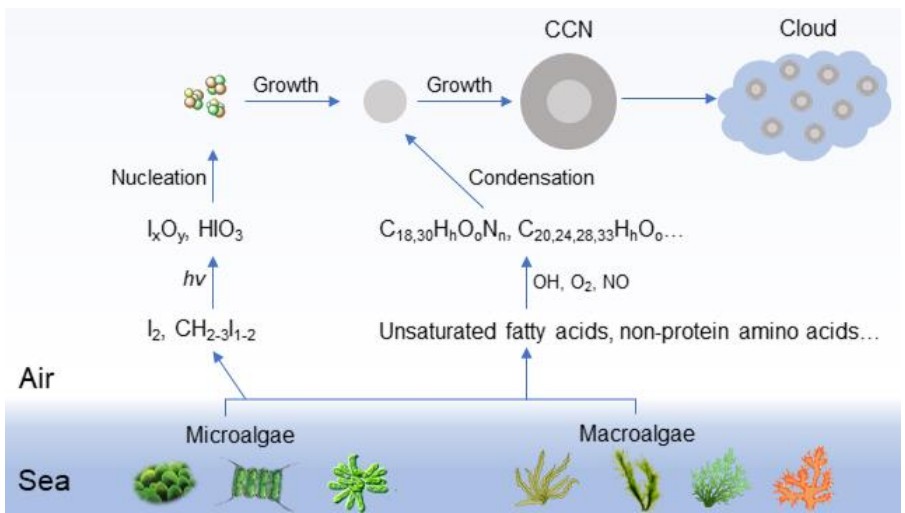

**ABSTRACT**
Unlike the deep understanding of highly oxygenated organic molecules (HOMs)
driving continental new particle formation (NPF), little is known about the organics
involved in coastal and open ocean NPF. On the coastline of China we observed intense
coastal NPF events initiated by iodine nucleation, but particle growth to cloud
condensation nuclei (CCN) sizes was dominated by organic compounds. This article
revealed a new group of $C_{18,30}H_hO_oN_n$ and $C_{20,24,28,33}H_hO_o$ compounds with specific
double bond equivalents and oxygen atom numbers in sub-20 nm coastal iodine new
particles by using ultrahigh resolution Fourier transform-ion cyclotron resonance mass
spectrometry (FT-ICR-MS). We proposed these compounds are oxygenated or nitrated
products of long chain unsaturated fatty acids, fatty alcohols, non-protein amino acids
or amino alcohols emitted mutually with iodine from coastal biota or biological-active
sea surface. Group contribution method estimated that the addition of $-ONO_2$, $-OH$ and
$-C=O$ groups to the precursors reduced their volatility of by 2~7 orders of magnitude
and thus made their products condensable onto iodine new particles in the coastal
atmosphere. Non-target MS analysis also provided a list of 440 formulas of iodinated



organic compounds in size-resolved aerosol samples during the iodine NPF days, which
facilitates the understanding of unknown aerosol chemistry of iodine.

## 1. INTRODUCTION

Atmospheric new particle formation (NPF) contributes over half of global cloud
condensation nuclei (CCN) (Merikanto et al., 2009) and thus influences cloud
properties and Earth's radiation budget (Metzger et al., 2010). By deploying high
resolution Chemical Ionization Mass spectrometer, recent laboratory and field studies
have identified a group of highly-oxidized organic molecules (HOMs) with high O/C
ratio and extremely low volatility from the reactions of volatile organic compounds
(VOCs) such as monoterpenes (Ehn et al., 2014). Sesquiterpenes (Richters et al., 2016)
and alkene (Mentel et al., 2015) with hydroxyl radical (OH), ozone ($O_3$) and nitrate
radicals ($NO_3$). These HOMs play an important role in particle nucleation and growth
of continental NPF, as well as in the formation of secondary organic aerosols.
Unlike the deep understanding of continental HOMs, little is known about the role
of organics in the NPF in coastal or open ocean atmosphere. The current state of
knowledge is that the self-clustering of biogenic iodine oxides or oxoacids could initiate
NPF events with particle number concentration sometimes exceeding $10^6 \, cm^{-3}$ (O'Dowd
et al., 2002; Burkholder et al., 2004; Sipilä et al., 2016; Stevanović et al., 2019; Kumar
et al., 2018). But it is unknown if other species are needed to drive the growth of iodine
clusters to CCN sizes in coastal or open ocean atmosphere (Saiz-Lopez et al., 2012).
Iodine-induced NPF (I-NPF) events were mostly reported in European coastlines (Yoon
et al., 2006; Mahajan et al., 2010) and polar regions (Allan et al., 2015; Roscoe et al.,
2015; Dall´Osto et al., 2018). In 2019 we provided evidences of I-NPF in the southeast
coastline of China, based on particle number size distribution and iodine measurements
(Yu et al., 2019). The focus of that paper (Yu et al., 2019) is, however, the speciation
of organic iodine compounds in size-segregated aerosol samples. Moreover, the use of
relatively low resolution Time-of-Flight (TOF) mass analyzer and *in vitro* signal
amplification approach in that paper did not allow the detection of the majority of non-





aromatic organic iodine compounds. Organic iodine remains to be the most significant
unknown in aerosol iodine chemistry at present (Saiz-Lopez et al., 2012).
Fourier Transform Ion Cyclotron Resonance (FT-ICR) coupled with soft ionization
techniques such as electrospray ionization (ESI) and ambient pressure chemical
ionization (APCI) allows characterization of complex organic mixtures at the molecular
level due to its ultra-high resolution and mass accuracy (Pratt and Prather, 2012). This
technique has been used to examine molecular composition of organic aerosols (Schum
et al., 2018; An et al., 2019; Zuth et al., 2018; Daellenbach et al., 2018; Xie et al., 2020)
and cloud water (Zhao et al., 2013; Bianco et al., 2018). Studies investigating coastal
organic aerosols have been rarely. Virtually no study reported the characterization of
organic compounds driving the growth of coastal or open ocean new particles.
In this study, comprehensive chemical composition analyses were conducted on the
size-segregated aerosol samples down to 10 nm, collected by 13-stage nano-MOUDI
(nano-micro orifice uniform deposit impactor) during the intense I-NPF days at a
coastal site of China. Relative abundances of $HSO_4^-$, total iodine and total organic
carbon (TOC) in 10-56 nm particles were compared between the I-NPF days and
conventional continental NPF days. In particular, using ultra-high resolution FT-ICR
MS, we conducted a non-target analysis of particle-phase organic compounds to
explore their molecular identity, formation mechanism and the role in new particle
growth in the coastal atmosphere.
**2. METHDOLOGY**
**2.1. Sampling collection**
The sampling site (29°29′ N, 121°46′ E) is in a building about 40 and 200m away
from the coastline of East China Sea (Zhejiang Province) at high tide and low tide,
respectively. The classification of I-NPF event and continental regional NPF (C-NPF)
event was based on particle number size distributions (PNSD) between 2 and 750 nm
monitored from January to May 2018 by a scanning mobility particle spectrometer



(SMPS; TSI DMA3081 and CPC3775; scanning range: 40-750 nm) and a neutral
cluster air ion spectrometer (NAIS; scanning range: 2-42 nm). Strong I-NPF events
were observed almost every day in April and May, which was the growth and farming
season of seaweed. A nano-MOUDI sampling scheme was implemented according to
the PNSD measurement. One set of nano-MOUDI samples was collected during the C-
NPF days from February 11 to 13; a second set was collected during the non-NPF days
from April 16 to 18; a third set was collected during the I-NPF days from May 9 to 11.
The PNSD during the 3 periods are shown in Figure S1. Each set of nano-MOUDI
samples was collected continuously for 72 hours, during which NPF occurred on a daily
basis, so that particle chemical composition of different event types can be obtained
from offline analyses. Aluminum foil filters were used as sampling substrate to avoid
the adsorption of gaseous species. For each set of nano-MOUDI samples, two nano-
MOUDIs were placed side by side to collect 10-100 nm particles (on stages 10-13; other
stages were silicon greased) and 100 nm-18 μm particles (on stages 1-9) separately, in
order to reduce potential positive particle-bounce artifacts. Three additional sets of
blank samples were collected by placing a high efficiency particulate air (HEPA) filter
at the gas inlet of nano-MOUDI. Detailed information on aerosol sample collection
could be found in Yu et al. (2019).
**2.2. Sample preparation and analysis**
Half of each filter was transferred into a 20 mL tapered plastic centrifuge tube, added
10 mL mixed solvent (1:1 v/v water and methanol; LCMS grade). The mixture was
sonicated for 40 min and filtered by a 0.2 μm PTFE membrane syringe filter. The filtrate
was evaporated to almost dryness in a rotary evaporator below 40 °C and subsequently
redissolved in 0.5 mL water. After being centrifuged for 30 min at 12,000 rpm, the
supernatant was collected for total iodine (I) analysis by Agilent 7500a ICP-MS
(Agilent Technologies, Santa Clara, CA, USA) and $HSO_4^-$ analysis by UPLC-ESI-Q-
TOF-MS. The measurements of $HSO_4^-$ and total I were elaborated in our previous
article Yu et al. 2019. Another half of each filter was extracted in the same way but
used for TOC analysis by a TOC analyzer (Model TOC-5000A, Shimadzu, Japan) and



non-target MS analysis of organic compounds (OC) by ESI-FT-ICR-MS (SolariX XR
9.4T instrument, Bruker Daltonics, Coventry, UK). Samples were infused by a syringe
pump and analyzed in both positive (ESI+) and negative (ESI-) modes. ESI-FT-ICR
MS operation conditions are included in Supplement Material. Field blank sample
extracts were analyzed following the same procedure.

**2.3. FT-ICR MS data processing**

A resolving power ($m/\Delta m_{50\%}$) 550,000 at m/z 300 of our FT-ICR-MS allows the
determination of possible formulas for singly charged molecular ions. Only m/z values
between 150-1000 Da that satisfies signal/noise (S/N) ratio > 10 were considered. For
each m/z value, several scientific rules were applied to calculate a reasonable elemental
formula of natural organic molecule: the general formula is $C_{1-50}H_{1-100}O_{0-50}N_{0-10}I_{0-3}$ in
the ESI+ mode; elemental ratios H/C, O/C, and N/C are limited to 0.3-3, 0-3 and 0-1.3,
respectively. The general formula is $C_{1-50}H_{1-100}O_{1-50}N_{0-5}S_{0-2}I_{0-3}$ in the ESI- mode;
elemental ratios H/C, O/C, N/C and S/C are limited to 0.3-3, 0-3, 0-0.5 and 0-0.2,
respectively. Mass error must be smaller than 0.5 ppm. Formula containing C, H, O, N,
S and I isotopologues were removed from the formula lists. A formula with m/z > 500
was not reported if it did not belong to any $CH_2$ homologous series. For a formula
$C_cH_hO_oN_nS_sI_x$, double bond equivalents (DBE) defined as $DBE = \frac{2c+2-h+n-x}{2}$ was
required to be non-negative. Formula calculation was done following the same
procedure for the three field blank samples. All formulas found in the field blank
samples, regardless of peak intensity, were excluded from the formula lists of real
samples. Aromaticity index (AI) is calculated from $AI = \frac{DBE_{AI}}{C_{AI}} = \frac{1+c-o-s-0.5h}{c-o-s-n}$. If
$DBE_{AI} \leq 0$ or $C_{AI} \leq 0$, then AI = 0. A threshold value of AI $\geq$ 0.5 provides an
unambiguous minimum criterion for the presence of aromatic structure in a
molecule(Yassine et al., 2014).



**3. RESULTS AND DISCUSSION**
**3.1. Organics dominate the growth of new particles initiated by iodine nucleation**
We first compare relative concentrations of major aerosol components, that is, total
I, $HSO_4^-$ and TOC, in nano-meter new particles during different event days. Total I
(13.5 ng m$^{-3}$, Table 1) in 10-56 nm particles during the I-NPF days was 67 and 36 times
higher than those during the C-NPF days (0.2 ng m$^{-3}$) and non-event days (0.37 ng m$^-$
$^3$). In the same size range, $HSO_4^-$ concentration (0.2 μg m$^{-3}$) during the I-NPF days was
lower than that during the C-NPF days (0.5 μg m$^{-3}$). This clearly indicates that the NPF
events from May 9 to 11 was linked to iodine nucleation. Even so, aerosol mass in 10-
56 nm new particles during the I-NPF days was dominated by organics. We define the
mass fraction of organic mass (OM) in the particles as $(1.5m_{TOC})/(m_{Total\ I} + m_{HSO4^-} +$
$1.5m_{TOC}) \times 100\%$ by assuming a OM/TOC ratio of 1.5. Mass fractions of OM are 95%,
87% and 68%, respectively, in the size bins 10-18 nm, 18-32 nm and 32-56 nm during
the I-NPF days. Therefore, organic compounds are critical for I-NPF to contribute to
CCN. The main purpose of this article is to identify these organic compounds during
the I-NPF days. The OC composition during the C-NPF days is beyond the scope of
this article.

Table 1. Concentrations of Total iodine (I), $HSO_4^-$ and Total Organic Carbon (TOC)
in 3 size bins between 10-56 nm during the I-NPF, C-NPF and non-NPF days. For
simplicity, only the sum of three size bins are shown for the C-NPF and non-NPF
days. BDL=below detection limit.

| | I-NPF | | | C-NPF | non-NPF |
|---|---|---|---|---|---|
| | 10-18 nm | 18-32 nm | 32-56 nm | 10-56 nm | 10-56 nm |
| Total I (ng m$^{-3}$) | 2.3 | 6.2 | 5.0 | 0.20 | 0.37 |
| $HSO_4^-$ (μg m$^{-3}$) | 0.022 | 0.034 | 0.144 | 0.50 | BDL |
| TOC (μg m$^{-3}$) | 0.31 | 0.18 | 0.21 | 0.28 | BDL |





**3.2. Elemental composition of non-iodinated OC on the I-NPF days**
Non-target analysis of OC elemental composition was performed in detail on 10-18
nm, 32-56 nm, 180-560 nm and 3.2-5.6 μm particles during the I-NPF days. Elemental
formulas in the 4 size bins can represent OC molecular composition of nucleation mode,
Aitken mode, accumulation mode and coarse mode, respectively. OC formulas were
divided into two categories: non-iodinated OC and iodinated OC. There are far more
non-iodinated OC formulas than iodinated OC formulas in <1 μm particles in terms of
both formula number (Table 2) and relative intensity (Figure 1). For example, 2831
non-iodinated OC formulas account for 96.6% of OC total intensity in 10-18 nm
particles, while 137 iodinated OC formulas account for the remaining 3.4%. It means
that non-iodinated OC dominates new particle growth during the I-NPF events. In this
section, we first discuss chemical characteristics of non-iodinated OC, while the
speciation of iodinated OC will be discussed in Section 3.4.

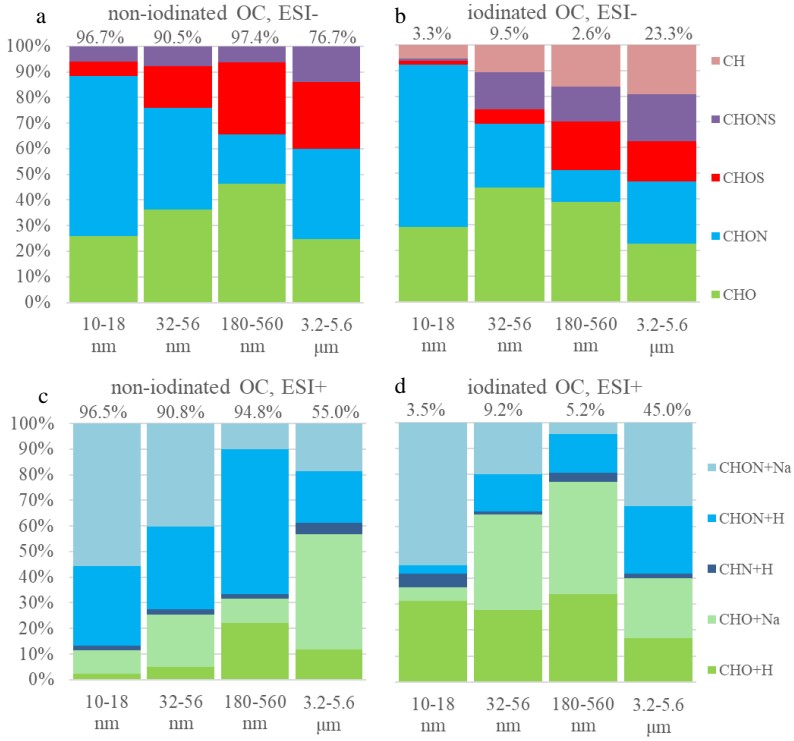






Figure 1. Relative intensity distributions of elemental groups observed in 10-18 nm, 32-
56 nm, 180-560 nm and 3.2-5.6 μm size bins in ESI+ and ESI- modes. The percentage
above a column denote the percent of non-iodinated OC (or iodinated OC) intensity in
total OC intensity in a size bin. +Na and +H denote $[M+Na]^+$ and $[M+H]^+$ adduct in
ESI+ mode, respectively.
The molecular formulas of non-iodinated OC were divided into seven elemental
groups $CHO^-$, $CHO^+$, $CHON^-$, $CHON^+$, $CHOS^-$, $CHONS^-$ and $CHN^+$. The number
distribution of 7 elemental groups for the 4 size bins is listed in Table 2. If both $[M+Na]^+$
and $[M+H]^+$ adducts of a formula were detected, the formula was counted only once. It
should be noted that some formulas were repeatedly detected in ESI+ and ESI- modes.
Some formulas detected in one size bin were also detected in another size bin. This is
quantitatively shown in the first four rows of Table 2. For instance, 58%, 25% and 4%
of the formulas detected in 10-18 nm aerosols were also detected in 32-56 nm, 180-560
nm and 3.2-5.6 μm aerosols, respectively. In another word, the particles in neighboring
size bins share more similarity in organic composition. An unexpected finding is that
the number of non-iodinated OC formulas detected in 3.2-5.6 μm coarse particles (n =
266) is one order of magnitude lower than those of other bins. Reconstructed mass
spectra of the 7 elemental groups in ESI-and ESI+ modes are shown in Figure S2 for
the four size bins.

Table 2. The numbers of assigned formulas of elemental groups of organic compounds
in 10-18 nm, 32-56 nm, 180-560 nm and 3.2-5.6 μm size bins. The first 4 rows show
the percent of formula repeatability between two size bins. 1I-OC: molecular formula
containing one iodine atom.

| Repeatability | 10-18 nm | 32-56 nm | 180-560 nm | 3.2-5.6 μm | |
|---|---|---|---|---|---|
| 10-18 nm | | 58% | 25% | 4% | |
| 32-56 nm | 57% | | 38% | 4% | |
| 180-560 nm | 34% | 51% | | 6% | |
| 3.2-5.6 μm | 35% | 35% | 34% | | |
| Non-iodinated OC | | | | | Total |
| $CHO^-$ | 531 | 565 | 525 | 20 | 892 |
| $CHO^+$ | 250 | 501 | 380 | 111 | 857 |





| | | | | | Total | 1I-OC (%) |
|---|---|---|---|---|---|---|
| CHON⁻ | 1005 | 638 | 347 | 25 | 1268 | |
| CHON⁺ | 1139 | 1055 | 828 | 72 | 2121 | |
| CHOS⁻ | 147 | 216 | 256 | 22 | 357 | |
| CHONS⁻ | 134 | 131 | 93 | 10 | 259 | |
| CHN⁺ | 34 | 26 | 7 | 7 | 46 | |
| Total | 2831 | 2770 | 2151 | 266 | 4979 | |
| Iodinated OC | | | | | Total | 1I-OC (%) |
| CHOI⁻ | 32 | 53 | 11 | 5 | 80 | 64% |
| CHOI⁺ | 17 | 85 | 31 | 31 | 136 | 93% |
| CHONI⁻ | 52 | 29 | 7 | 7 | 77 | 88% |
| CHONI⁺ | 34 | 57 | 18 | 52 | 132 | 81% |
| CHOSI⁻ | 3 | 8 | 7 | 3 | 18 | 72% |
| CHONSI⁻ | 2 | 7 | 3 | 2 | 13 | 62% |
| CHNI⁺ | 6 | 4 | 4 | 3 | 16 | 56% |
| CHI⁻ | 4 | 2 | 1 | 4 | 9 | 67% |
| Total | 137 | 228 | 76 | 100 | 440 | 80% |

CHON is the most commonly assigned elemental group in both ESI+ (2121 CHON⁺)
and ESI- (1268 CHON⁻) modes, followed by the CHO group (857 CHO⁺ formulas and
892 CHO⁻ formulas). S-containing formulas are 357 CHOS⁻ and 259 CHONS⁻. The
formula number of the least common CHN⁺ group is only 46. In terms of relative
intensity, CHON fraction in the ESI- mode decreases from 61% of OC in the 10-18 nm
bin to 20% in the 180-560 nm bin (Figure 1a), while the fractions of CHO and
CHOS/CHONS increase with particle size. In the ESI+ mode, the fraction of CHON
decreases from 88% in 10-18 nm bin to 70% in 180-560 nm bin, being always the
dominant elemental group of non-iodinated OC (Figure 1b). Low molecular weight
(LMW) amines are important stabilizers in acid-base nucleation (Kurtén et al., 2008;
Jen et al., 2014; Zheng et al., 2000; Yao et al., 2016), but their molecular ions are out
of the mass range of our FT-ICR-MS. The CHN⁺ formulas that we observed contained
9-50 C atoms and 1-7 N atoms, accounting for a negligible fraction 1.7% of total
intensity of all ESI+ formulas in the 10-18 nm particles.
Previous elemental composition studies using FT-ICR-MS were mostly conducted
on $PM_{2.5}$ or $PM_{10}$ collected from marine (Schmitt-Kopplin et al., 2012; Bao et al., 2018;
Ning et al., 2019), urban (Wu et al., 2019; Jiang et al., 2016), troposphere (Schum et al.,
2018; Mazzoleni et al., 2012) and forest sites (Kourtchev et al., 2013). In general, these
studies showed that the numbers of CHO compounds were comparable with or more





than those of CHON compounds. Our study shows clearly that elemental composition
of aerosol OC is highly size dependent. New particle growth in the size range of 10-18
nm during the I-NPF event is dominated by CHON elemental group, followed by CHO.
The focus of this article narrows on the identity and source of the CHON and CHO
formulas in 10-18 nm particles, by comparing with those in the 180-560 nm size bin.

### 3.2.1. CHO formulas

There is a total of 531 CHO⁻ formulas and 250 CHO⁺ formulas in 10-18 nm particles.
CHO formulas are commonly found in ESI+ and ESI- modes. In terms of relative
intensity, CHO⁻ compounds are more abundant than CHO⁺ compounds (Figure 3b, total
intensity: 4.14 e+09 *vs.* 1.24 e+09). However, this is not indicative of absolute
concentration of the two groups due to different ionization efficiency between ESI- and
ESI+ modes. CHO⁻ is characterized by a series of formulas with 20, 24, 28, and 33 C
atoms, 4 or 6 O atoms and 1 equivalent double bond (Figure 2b). The total intensity of
top 10 formulas accounts for 30% of all 531 formulas. Assuming CHO⁻ formulas
contain at least 1 carboxylic group (–COOH), the rest of their molecules should be
saturated (DBE = 0) and contain 2 or 4 O atoms.

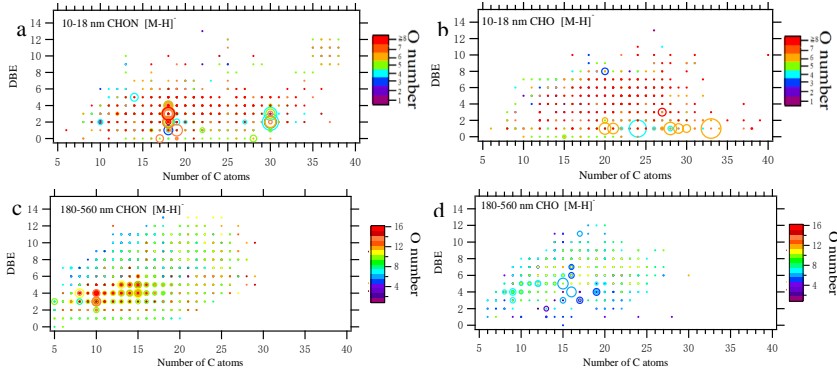


Figure 2. DBE *vs.* C atom number diagrams of all CHON and CHO formulas detected
in 10-18 nm and 180-560 nm particles in ESI- mode. The color bar denotes O atom





number in the formulas. The size of the circles reflects the relative intensities of
molecular formulas on a logarithmic scale.

The above feature is not seen in either CHO+ formulas in the 10-18 nm bin or CHO-
formulas in the 180-560 nm bin. There are more $C_{21}$ and $C_{24}$ formulas than other C
subgroups in the CHO+ formulas of 10-18 nm bin (Figure S3d), but none of them have
exceptionally-high intensity. The prominent formulas in the CHO- group in 180-560
nm particles have relatively high unsaturation degree (DBE = 3-7, Figure 2d). The
relative intensities of subgroups according to C atom number in the CHO- formulas in
the 180-560 nm bin are characterized by bimodal distribution with maximum intensity
around $C_{15}$-$C_{16}$ and $C_{20}$ (Figure 3d). The relative intensity of O atom subgroups is
mono-modally distributed around $O_7$ (Figure S4).

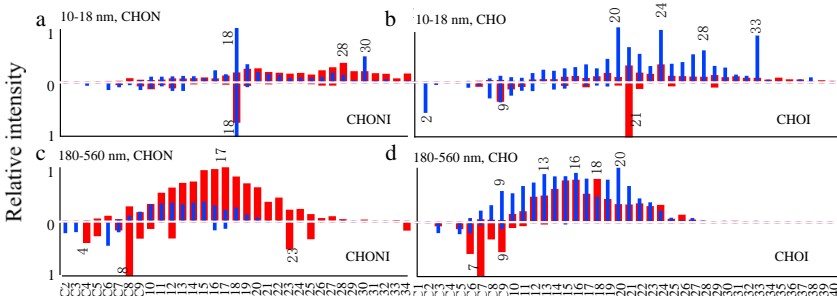


Figure 3. Relative intensities of subgroups according to C atom number in CHON, CHO,
CHONI and CHOI formulas in 10-18 nm and 180-560 nm particles in ESI+ (in red) and
ESI- (in blue). The intensity of the most abundant subgroup in a size bin is defined as
1 and those of other subgroups are normalized by it. The relative intensities of non-
iodinated OC formulas (iodinated OC formulas) are plotted in the region above (below)
zero line.

***3.2.2. CHON formulas***
As discussed earlier, CHON is the most abundant elemental group observed in the
smallest size bin 10-18 nm. There is a total of 1005 CHON- formulas (total intensity



9.96 e+09) and 1139 $CHON^+$ formulas (6.45 e+09) in 10-18 nm bin. 355 CHON
formulas are commonly found in ESI+ and ESI- modes. A close examination of Figure
2a and 3a reveals that $CHON^-$ is characterized by a series of $C_{18}$ and $C_{30}$ formulas with
low DBE values (1-4). 87 $C_{18}$ and 26 $C_{30}$ formulas account for 37% of total intensity of
$CHON^-$. Such feature is not seen for $CHON^+$ formulas that are rather uniformly
distributed in DBE vs. C diagram (Figure S3a and S3c). Generally speaking, $CHON^-$
compounds should contain nitro- ($–NO_2$) or nitrooxy- ($–ONO_2$) group and are ionizable
due to the presence of –COOH or hydroxy (–OH) (Lin et al., 2012). However, the
presence of amine group in $CHON^-$ formulas cannot be excluded. Take $C_{18}$ as example,
51 out of 87 $C_{18}H_hO_oN_n^-$ formulas should contain at least one amine group, either
because their O atom numbers are not large enough to allow the assignment of $–NO_2$
for all N atoms, or because some formulas (25 out of 87) were also detected in ESI+
mode. In total, 51 $C_{18}H_hO_oN_n^-$ formulas with an amine group account for 54.4% of total
intensity of 87 $C_{18}H_hO_oN_n^-$ formulas.
The presence of amine group in $C_{18}H_hO_oN_n^-$ formulas in 10-18 nm particles is also
supported by the comparison with $CHON^-$ in 180-560 nm submicron aerosols. $CHON^-$
in 180-560 nm is characterized by a number of formulas with maximum intensity
around $C_{10}$ and $C_{15}$ (Figure 2c). A plot of O atom number *vs.* N atom number in Figure
S5a shows that $C_{10}H_hO_oN_n^-$ in 180-560 nm have O/N ratios ≥ 3 and O atom number is
positively correlated with N atom number. It indicates that these $C_{10}H_hO_oN_n^-$ formulas
are probably nitro- or nitrooxy- oxidation products of monoterpenes from continental
plant emission. In contrast, O/N ratios of the $C_{18}H_hO_oN_n^-$ formulas in 10-18 nm are
mostly small and O atom number do not increase with N atom number (Figure S5b).
All collective evidences above reveal that nitrogen-containing organic compounds in
10-18 nm particles during the I-NPF days are partly composed of long-chain amino
alcohols, amino acids and so on.
In summary, a series of very distinctive $CHON^-$ and $CHO^-$ formulas was observed in
10-18 nm new particles during the I-NPF days. These formulas are characterized by
some specific numbers of C atoms (i.e. $C_{18}H_hO_oN_n$, $C_{30}H_hO_oN_n$, $C_{20}H_hO_o$, $C_{24}H_hO_o$,
$C_{28}H_hO_o$ and $C_{33}H_hO_o$) and equivalent double bonds (DBE = 1 for $CHO^-$ and 1-4 for





$CHON^-$). To the best of our knowledge, such $CHON^-$ and $CHO^-$ formulas have not been
reported by previous aerosol studies. The chemical composition of new particles is
completely decoupled with the $CHO^-$ and $CHON^-$ formulas around $C_{10}$, $C_{15}$ and $C_{20}$ in
180-560 nm submicron particles, which might be originated from continental terpene
emissions. Fewer O atoms in $C_{18,30}H_hO_oN_n$ and $C_{20,24,28,33}H_hO_o$ formulas than those in
submicron aerosols indicate that these compounds should be more freshly emitted into
the atmosphere. The discontinuous chemical composition and PNSD spectrum (Figure
S1a) below and above 50 nm particle size reflect the fact that the further growth of new
particles beyond 50 nm in local I-NPF events cannot be monitored by our stationary
sampling strategy.
On the other hand, we observed more complicated distributions of $CHO^+$ and
$CHON^+$ formulas in 10-18 nm new particles that are of relatively small individual
intensity and are rather uniformly distributed in DBE vs. C diagrams. Like $CHON^-$ and
$CHO^-$, those $CHO^+$ and $CHON^+$ formulas also possess a larger number of C atoms (C >
19) than their counterparts in 180-560 nm submicron aerosols (Figure 3). 21 out of 30
most abundant $CHON^+$ formulas contain two or more N atoms; this ratio 21/30 is higher
than those in $CHON^-$ formulas. Generally speaking, $CHO^+$ and $CHON^+$ formulas
represent carbonyls/alcohols/epoxides and amino alcohols/amino acids, respectively.
One interesting finding about $CHO^+$ and $CHON^+$ is that they tend to form $[M+Na]^+$
adducts in small aerosols and $[M+H]^+$ adducts in large aerosols (Figure 1c). This
indicates that the $CHO^+$ and $CHON^+$ compounds in new particles during the I-NPF days
should possess different basic functional groups from those in submicron particles.
**3.3. Possible precursors and formation mechanism of organic compounds**
**detected in 10-18 nm new particles during the I-NPF days**
It is unrealistic to simply propose one out of a large number of possible structures for
a formula with large C atom number (e.g., $\geq 18$). Our strategy is to first explore the
possible precursors of the newly found CHON and CHO formulas. It is obvious that
these $C_{18,30}H_hO_oN_n$ and $C_{20,24,28,33}H_hO_o$ formulas cannot be attributed to continental





terpene emission or anthropogenic aromatic emissions. Previous field measurements of
marine NPF precursor focused on volatile species like iodine (Stevanović et al., 2019),
iodomethanes (O'Dowd et al., 2002), dimethyl sulfonic acid (Yvon et al., 1996; Barone
et al., 1996; Barnes et al., 2006) and LMW amines (Ning et al., 2019; Ge et al., 2011).
So far there is no report about aliphatic compounds with C number $\geq 18$ in either gas
phase or new particles (Cochran et al., 2017; Bikkina et al., 2019). Therefore, we
consulted the literature that reported chemical compounds isolated from biological
tissues of algae, plankton or other marine organisms. Potential precursors are listed in
Table 3.

### 3.3.1.  *Fatty acids*

Fatty acids (FAs) are widely found in animals, plants and microbe (Moss et al., 1995).

Plants have higher content of unsaturated FAs than animals. $C_{14}$-$C_{24}$ fatty acids,
including both saturated and unsaturated, have long been observed in seaweed
(Dawczynski et al., 2007) and very long chain FAs ($C_{24}$-$C_{36}$) have been isolated from
green algae, chlorella kessleri, sponges and marine dinoflagellate (Litchfield et al., 1976;
Řezanka and Podojil, 1984; Mansour et al., 1999). $C_{18}$ Oleic acid, linoleic acid and
linolenic acid are most commonly found unsaturated FAs in macro algae. FAs with
carbon chain shorter than $C_{20}$ were used by atmospheric chemists as organic tracers of
atmospheric aerosols from microbe or kitchen emission (Simoneit and Mazurek, 1982;
Zheng et al., 2000; Guo et al., 2003; Rogge et al., 1991; DeMott et al., 2018;
Willoughby et al., 2016). In our study, no saturated FAs were detected in 10-18 nm
particles. Only 1.5% CHO⁻ formulas can be assigned to unsaturated FAs (that is, include
2 O atoms, 14-28 C atoms and DBE = 3-6). Other CHO compounds observed in 10-18
nm particles contain > 2 O atoms and thus can be assigned as the oxidized derivatives
of FAs.

Table 3 Possible precursors and their presence in marine biological sources and our
aerosol samples. ND: not detected.

| Potential precursors | Presence in marine sources | Presence in aerosol particles |
| --- | --- | --- |



| Unsaturated fatty acid | $C_{14}$-$C_{24}$ fatty acids | Seaweed (Dawczynski et al., 2007) | 1.5% of CHO⁻ in terms of relative intensity |
|---|---|---|---|
| | $C_{25}$--$C_{36}$ very long chain fatty acids | Green algae, chlorella kessleri, sponges, marine dinoflagellate (Litchfield et al., 1976; Řezanka and Podojil, 1984; Mansour et al., 1999). | ND |
| fatty alcohols | $C_{30}$-$C_{32}$ mono- and diunsaturated alcohols and diols | Yellow-green algae (Volkman et al., 1992)(eustigmatophytes) | ND |
| Saturated hydroxyl fatty acids | $C_{20}H_{40}O_3$, $C_{32}H_{64}O_4$ | Nannochloropsis, cutins and suberins of higher plants (Gelin et al., 1997). | S/N 15 and 28 |
| Nonprotein amino acid | $C_{18}H_{37}NO_4$ saturated dihydroxy amino acid (simplifungin, $C_{20-22}H_{39-41}NO_{5-7}$ mono-unsaturated polyhydroxy amino acids in sphigolipids | Marine fungal metabolites (Ishijima et al., 2016; VanMiddlesworth et al., 1992). | S/N 280 S/N 30-230 |
| Amino alcohols | $C_{16-28}H_{33-53}NO_{1-4}$ polyhydroxy amino alcohols | Plant biomembrane, secondary metabolites in marine organisms (Jares-Erijman et al., 1993). | S/N 23-640 |
| | $C_{18}H_{31}NO$ and $C_{18}H_{29}NO$ polyunsaturated amino alcohols | Mediterranean tunicate (Jares-Erijman et al., 1993) | S/N 10-60 |
| | $C_{18}H_{36}N_2O_5$ polyhydroxy cyclic alkaloid | Moraceae (Tsukamoto et al., 2001) | S/N 800 |


Possible oxidation schemes of two typical $C_{18}$ ($C_{18}H_{30}O_2$, α-linolenic acid, three C=C
double bonds in carbon chain) and $C_{28}$ unsaturated FAs ($C_{28}H_{52}O_2$, two C=C double
bonds), for instance, are proposed in Figure S6 and S7. The reaction of an unsaturated
FA after the emission into the atmosphere is initiated by OH addition to C=C double
bond and subsequent $O_2$ addition to form a peroxy radical (Atkinson et al., 1995;
Calvert et al., 2000). Depending on the level of NO and reactivity, four competitive
pathways are available for peroxy radicals to produce CHO or CHON formulas
observed in our study: reaction with NO to form a –$ONO_2$ group (pathway 1) or an
alkoxy radical that further reacts with $O_2$ to form a carbonyl (–C=O, pathway 2),
reaction with $RO_2$ radicals to form a hydroxyl (–OH) or a –C=O group (pathway 3) and
successive intermolecular H-shift/$O_2$ addition autoxidation(Crounse et al., 2013;
Vereecken et al., 2015) (pathway 4).
We propose that pathway 1 and 2 are preferred for those FAs (e.g., α-linolenic acid)
with higher reactivity with NO (Figure S6). Pathways 1 and 2 add –$ONO_2$, –OH and –
C=O groups to carbon chain but do not reduce the DBE of FA precursor. α-linolenic
acid oxidation in the atmosphere via sequential occurrences of pathways 1 or 2 yields



a series of oxygenated and nitrated organic compounds, among which $C_{18}H_{31}NO_6$,
$C_{18}H_{31}NO_8$, $C_{18}H_{31}NO_{10}$, $C_{18}H_{32}N_2O_{10}$ and $C_{18}H_{33}N_3O_4$ are found in 10-18 nm particles.
These formulas explain the circles with DBE = 4 and C number = 18 shown in Figure
2a DBE *vs.* C atom number diagram.

We propose that pathway 3 and 4 are preferred for those FAs (e.g. $C_{28}$ FA $C_{28}H_{52}O_2$)

with higher reactivity with $RO_2$ (Figure S7). The net outcome of sequential pathway 3
and 4 reactions is to add –OH and –C=O groups and reduce the DBE of FA precursor.
The end products are a series of $C_{28}H_{52}O_{6-8}$, $C_{28}H_{54}O_{4-7}$ and $C_{28}H_{56}O_{6-8}$ compounds,
which can explain the circles with C number = 28 and DBE = 1-3 in Figure 2b.

In addition to fatty acids, fatty alcohols such as $C_{30}$-$C_{32}$ mono- and di-unsaturated

alcohols and diols have been detected in yellow-green algae (Volkman et al., 1992).
Although these unsaturated alcohols were not detected in our 10-18 nm particles, we
suppose that they or their metabolites in algae may undergo similar reactions like
unsaturated FA to generate condensable oxygenated and nitrated fatty alcohols in the
atmosphere. Hydroxy fatty acids (HFAs) are important constituents of lipid in marine
microalgae (Gelin et al., 1997), bacteria (Kim and Oh, 2013),seaweed (Kendel et al.,
2013; Blokker et al., 1998) and leaf surface of higher plants (Pollard et al., 2008).
Among them, two saturated HFAs $C_{20}H_{40}O_3$ and $C_{32}H_{64}O_4$ were found in our 10-18 nm
aerosol sample with S/N ratios 15 and 28. However, because saturated hydroxy fatty
acids are not oxidizable via the pathways proposed in our study, they are assumed
unlikely to be precursors of other formulas observed in 10-18 nm particles.
### 3.3.2.   *Nonprotein amino acids and amino alcohols*

Quantum chemical calculations have showed that amino acids like Glycine, Serine,

and Threonine are potential participants in atmospheric nucleation via interaction with
sulfuric acid (Elm et al., 2013; Ge et al., 2018; Li et al., 2020). However, we did not
observe any of 20 essential amino acids in 10-18 nm in either ESI+ or ESI- modes. One
reason may be that most of essential amino acids have molecular weight less than 150
that is below mass scan range of our FT-ICR-MS.

There are a number of records in the literature about long chain non-protein amino

acids or amino alcohols isolated from marine organisms or plant biomembrane





(Ishijima et al., 2016; VanMiddlesworth et al., 1992; Jares-Erijman et al., 1993;
Tsukamoto et al., 2001). They include saturated dihydroxy amino acid ($C_{18}H_{37}NO_4$,
DBE = 1, simplifungin), monounsaturated polyhydroxy amino acids in sphigolipids
($C_{20-22}H_{39-41}NO_{5-7}$, DBE = 2-3), polyhydroxy amino alcohols ($C_{16-28}H_{33-53}N_{1-2}O_{1-5}$,
DBE = 1-3, sphingosine and its natural metabolites) and polyunsaturated amino
alcohols ($C_{18}H_{31}NO$ and $C_{18}H_{29}NO$, DBE = 4-5). All of these formulas were detected
in 10-18 nm aerosols with S/N in the range of 10-800. More importantly, all those
compounds that contain at least one amine group and one C=C double bond can be
precursors of observed CHON formulas containing amine group via the pathways that
we showed above. As an example, the oxidation scheme of an amino alcohol $C_{18}H_{31}NO$
with 4 C=C double bonds in carbon chain is illustrated in Figure S8.
Similar to $C_{28}$ FA, $C_{18}H_{31}NO$ undergoes successive intermolecular H-shift/$O_2$
additions to produce a series of $RO_2$ radicals with hydroperoxyl group (–OOH) in its
carbon chain. The subsequent pathway 3 reactions, as well as the decomposition of –
OOH groups, add –OH and –C=O groups in the carbon chain. Because $C_{18}H_{31}NO$
possesses as many as 4 C=C double bonds, sequential pathway 3 and 4 reactions
produce a large number of oxidation products, among which 57 are found in the formula
list detected in 10-18 nm particles (Figure S8). These products $C_{18}H_{31}NO_{4-11,13}$,
$C_{18}H_{33}NO_{4,6-10}$, $C_{18}H_{35}NO_{5-9}$, $C_{18}H_{37}NO_{7-12}$ and $C_{18}H_{39}NO_{10-11}$ explain perfectly the
presence of a series of formulas with C number = 18, DBE = 0-4 and a –$NH_2$ group
shown in Figure 2a.

### 3.3.3. Volatility estimation

Based on the reaction mechanisms proposed above, it is possible to estimate the
volatility change from potential precursors to their oxidation products. First, from the
list of elemental formulas detected in 10-18 nm particles, we select 49 formulas with
high intensities, including 14 $CHON^-$ formulas with peak intensity > 1.00 e+08, 23
$CHON^+$ formulas with peak intensity > 3.00 e+07 and 12 $CHO^-$ formulas (DBE = 1)
with peak intensities > 3.00 e+07. Possible combinations of –COOH, –$ONO_2$, –C=O,
C=C double bond, –$NH_2$ and –OH groups are searched for every formula obeying two
simple rules: $CHON^-$ and $CHO^-$ formulas must possess a carboxyl or hydroxyl group;





$CHON^+$ formulas must possess an amino group. Saturation concentrations (C*) of the
49 formulas were then predicted for all combinations using a simple group contribution
method developed by Pankow and Asher (Pankow and Asher, 2008). On the other hand,
the C* of their possible precursors, including unsaturated FAs, fatty alcohols,
nonprotein amino acids or amino alcohols, were predicted by the same method.
As we can see in Table S4, C* of the 49 formulas fall into the range of ELVOC ($<$ 3
$\times\ 10^{-5}$ µg m$^{-3}$), while C* of their precursors are in the range of SVOC (0.3-300 µg m$^{-3}$)
or LVOC ($3 \times 10^{-5}$-0.3 µg m$^{-3}$). The addition of functional groups reduces the volatility
of precursors by 2~7 orders of magnitude and thus make their oxidation products
condensable onto new particles during the I-NPF event days. Therefore, the analysis of
precursor-product volatility partly supports our hypothesis about the molecular identity
and formation mechanism of the formulas detected in 10-18 nm particles.
**3.4. Speciation of iodinated OC**
Organic iodine compounds hold the key to understand aerosol iodine chemistry and
its role in regulating the recycling of halogens to the gas phase. We identified 440
iodinated OC formulas from the 4 size bins during the I-NPF days (Table 2). 80% of
the 440 formulas contain one I atom and the rest of them contain two I atoms. In terms
of relative intensity, iodinated OC accounts for 2.6-9.5% of OC in fine particles, but its
fraction in coarse particles increases to 23.3% in ESI- mode and 45% in ESI+ mode.
The size distribution of 7 iodinated OC groups (i.e., $CHOI^-$, $CHONI^-$, $CHOSI^-$,
$CHONSI^-$, $CHOI^+$, $CHONI^+$ and $CHNI^+$) resembles those of non-iodinated OC groups
(Figure 1). If we replace I atom(s) with H atom(s) in a formula, 107 out of 440 replaced
formulas are also found in the non-iodine OC formula list.



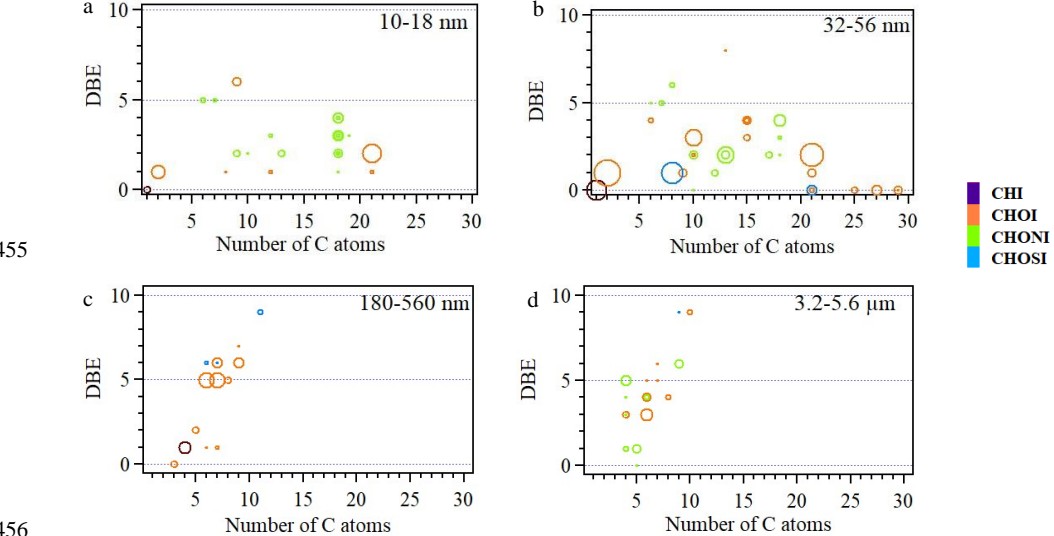

Figure 4. DBE vs. C atom number diagrams of iodinated OC formulas with intensity > 1.00 e+07 in the four size bins. The color bar denotes the elemental groups of assigned formulas. The size of the circles reflects the relative intensities of molecular formulas on a logarithmic scale.

Iodinated OC with intensity > 1.00 e+07 in the four size bins were shown in Figure 4. The DBE vs. C diagram for 10-18 nm particles is characterized by (1) nine $C_{18}H_hO_oN_nI$ formulas with DBE = 1-4 and (2) $C_9H_{16}NO_3I$ and its $C_{10}$-$C_{13}$ homologues. Because these formulas were detected in ESI+ mode, they are most likely iodinated amino acids. 32-56 nm particles accommodate most abundant iodinated OC formulas, in terms of both formula number and relative intensity. Prominent formulas include (1) diiodo acetic acid $C_2H_2O_2I_2$, diiodomethane $CH_2I_2$, (2) iodinated $C_{21}$ carbonyls $C_{21}H_{39}OI$ and $C_{21}H_{41}OI$, (3) iodinated $C_{21,25,27,29}$ alcohols or ethers with DBE = 0, (4) iodinated $C_{10}$ and $C_{15}$ terpene and sesquiterpene oxidation products and (5) iodinated organic sulfate $C_8H_{17}N_2SO_8I$ and $C_{21}H_{43}SO_4I$. In addition, $C_9H_{10}NO_3I$ detected in this size bin (S/N ratio: 35) can be tentatively assigned to an iodinated amino acid iodotyrosine that has been observed in seaweed(Yang et al., 2016), implying direct contribution from seaweed emission to new particles.



In 180-560 nm particles, the majority of iodinated OC are $C_{6-9}$ aromatic $CHOI^+$
compounds with AI > 0.5 and DBE = 5-7. Both C and O atom numbers of these
iodinated OC are smaller than those of mono-modally distributed $CHO^+$ compounds
around $C_{15}$ in the same particle size (Figure 3d and S3b). This implies that iodine has a
strong tendency to aromatic compounds in submicron aerosols due to electrophilic
substitution on aromatic rings. In 3.2-5.6 μm particles, iodinated OC features $C_4$-$C_6$
$CHO^+$ and $CHON^+$ compounds with DBE = 3-6, which again have fewer C atoms than
non-iodinated OC. Supporting evidence from AI shows these compounds are not
aromatic. Coastal 3.2-5.6 μm particles can be sea salt particles formed during bubble
bursting of sea water (Russell et al., 2010; Schmitt-Kopplin et al., 2012; Quinn et al.,
2014; Wilson et al., 2015). However, Hao et al. 2017 (Hao et al., 2017) showed that
iodinated OC products from the reaction between iodine and seawater are highly
unsaturated carboxylic-rich polyphenols with DBE = 4-14 and C atoms = 10-30. It is
thus apparent that iodinated OC in 3.2-5.6 μm particles were not directly from iodinated
organic compounds in the seawater.
**3.5 Atmosphere implications**
Due to the 71% ocean coverage of the earth's surface, marine aerosol generation is
important in determining the earth's radiative budget and climate change. Of the limited
number of studies reporting coastal NPF, most have focused on iodine emission,
oxidation and nucleation in the early stage of NPF. In principle, abundant low-volatility
condensing vapors other than iodine are required in coastal environments for the growth
of iodine clusters to CCN. This article reveals a new group of important organics
involved in this process. It is most likely that their precursors are emitted mutually with
iodine from either direct exposure of coastal biota to the atmosphere or biological-active
sea surface. More fundamental field, laboratory and modeling studies are needed to
determine (1) exact emission sources and source rates of these organic precursors, (2)
their gas phase intermediates and oxidation mechanisms in the atmosphere and (3) their
quantitative contribution to global and regional CCN numbers.





**ACKNOWLEDGMENTS**


The work was supported by the National Science Foundation of China (grant
numbers 41975831 and 41675124) and the National Key Research and Development
Program of China (grant number 2016YFC0203100).

***Data availability.*** The data used in this publication are available from the
corresponding author Huan Yu (yuhuan@cug.edu.cn).

***Author contributions.*** Huan Yu designed and conducted chemical analysis. Yibei
Wan and Huan Yu did data analysis and wrote the paper. Xiangpeng Huang
conducted the field sampling. Bin Jiang and Yuhong Liao did the FT-ICR-MS
analysis. Binyu Kuang, Manfei Lin, Deming Xia, Jingwen Chen and Jianzhen Yu
reviewed and revised the manuscript.

***Conflict of Interest***
The authors declare that they have no conflict of interest.

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
