# Peer review of "iodine nucleation in coastal atmosphere"

_Atmospheric Chemistry and Physics, 2020_

## Referee Comment (RC1) · Anonymous Referee #1 · 19 May 2020

Review of Wan et al.:

The paper presents a thorough report on the observation of a new group of aerosol organic species in coastal new iodine particles. Further effort is directed towards identifying the nature of such organic compounds, and their different contribution to size-resolved aerosol samples. The paper is very detailed and very well written, although some language editing may be beneficial. This paper fits well within the scope of ACP. I would like to congratulate the authors for this novel iodine research and I recommend publication after the comments below are addressed:

-   The authors argue the source of the reported organics is likely the same as for iodine emissions (i.e. algae exposed to the atmosphere at low tide). However, they should elaborate more about this since the results shown in the paper are not clear on this regard. For instance, do they see similar levels of OC at high tide when no iodine emissions occur?; have the authors conducted air mass back trajectories to track the origin and exposure levels of the samples air masses?; is there any anthropogenic influence on the levels of OC measured in iodine particles?. Overall, the paper would benefit from a some more detailed discussion on the possible sources of OC and the potential anthropogenic influence. This is important to be able to extrapolate their results to other iodine-rich coastal locations.
-   At low tide, the instrument is between 40 and 200 m from the coastal line and the emission area. Can the authors comment on the effect of this distance on their measurements?, can they see gradual differences in aerosols composition as the water recedes?
-   Line 49, beginning of paragraph. It would be useful for the reader to clarify that organic iodine is not the main source of the iodine oxide precursors that lead to the formation of I-NPF. This was shown in the past for coastal regions:

    A.  Saiz-Lopez and J. M. C. Plane, Novel iodine chemistry in the marine boundary layer. *Geophys. Res. Lett.* **31**, L04112 (2004).

    and for opean ocean conditions:

    A.  S. Mahajan *et al.*, Measurement and modelling of tropospheric reactive halogen species over the tropical Atlantic Ocean. *Atmos. Chem. Phys.* **10**, 4611-4624 (2010).

    A.  S. Mahajan et al., Latitudinal distribution of reactive iodine in the Eastern Pacific and its link to open ocean sources, *Atmospheric Chemistry and Physics*, **12**, 11609-11617 (2012).

-   Final comment. I would suggest to add a sentence to the Conclusions about the potential relevance of extrapolating the results of this location to other coastal locations, based on the questions above.

---

## Referee Comment (RC2) · Anonymous Referee #2 · 2 Jun 2020

**Review of Wan et al., 2020:**

This paper contains reports on the multitude of organic compounds present in size segregated aerosol samples at a coastal location where frequent new particle formation (NPF) was observed, as measured by a powerful off-line mass spectrometric technique. Iodine-NPF events are identified (I-NPF) and these are linked to aerosol chemical composition during periods with three days of consecutive I-NPF days. These are compared with similar data for C-NPF days. The potential sources are discussed, and volatilities of these compounds as calculated by a group contribution method are presented and discussed with a view to linking these compounds to new particle growth. The paper is well within the remit of ACP and contains a very valuable dataset in a field where not much is yet known, however, I find issues with the authors' definition of I-NPF, and the discussion of calculated vapour pressures could do with some extension as there are many uncertainties with these calculations not discussed. I recommend publication after the following issues are addressed.

**General comments:**

- In **section 3.1** the authors present 10-56 nm particle phase composition in 72 hour aerosol samples through 3 consecutive NPF days of two different categories (iodine-induced NPF, or *I-NPF*, & continental regional NPF, or *C-NPF*), identified from size-distribution measurements. It is evident from the size distributions that these NPF days *are* very dissimilar, and the evolution of the size distribution is markedly similar to prior reports of iodine-nucleation (such as Sipilä et al., 2016), however, the authors state that an elevation to the mass concentration of iodine in 10-56 nm aerosol samples during the I-NPF days is clear indication that NPF was linked to iodine nucleation. I would strongly argue this is not a clear indication a nucleation process involving iodine vapours, as aerosol mass between 10-56 nm is not an indicator of what process produces particles at ~1.7 nm.

  I would urge the authors to back this claim up with reference to similar reports of iodine driven nucleation under conditions that produce either similar particle composition or similar size distributions. $HSO_4^-$ concentrations are still an order of magnitude higher than $I^-$ concentrations in 10-18 nm particles on I-NPF days. If there is there any evidence within the data that nucleation processes are not dominated, for example, by sulphuric acid processes, this would strengthen this section greatly.

- It would be very useful to mention the magnitude of the effect of these NPF events to aerosol mass in the relevant size ranges – coming up with an exact number is a rather uncertain process, but a simple alternative would be a plot showing the time evolution of size-segregated aerosol mass as currently there is no indication of how much of the aerosol mass is actually arising from NPF.

- Recent chamber results utilising EESI-ToF-MS would indicate that oligermisation processes in the particle phase can produce a diverse range of compounds (Pospisilova et al., 2020), as well as particle phase processes producing organosulphates (Mutzel et al., 2015), formation of ester hydroperoxides (Zhao et al., 2018) etc. I would consider how such mechanisms may affect your proposed formation mechanisms, and further your estimations of volatility - if some of these compounds arise from particle-phase oligomerisation, does this change the conclusions of the paper? I would like to echo reviewer #1 here and suggest an extension of the discussion so the results can be more easily extrapolated to other coastal regions.

- The estimation of volatilities through SIMPOL has some associated uncertainties, and, at least in the case of HOMs, estimations of volatility can vary by several orders of magnitude when compared with other methods (see Figure 8 of Peräkylä et al., 2019). This should be a point of discussion as it may affect the outcomes of the paper.

**Specific comments:**

Line 22 and throughout: Change "organics" to "organic compounds".

Line 43: "Highly oxygenated (multifunctional) organic molecules" is a more correct term for these than "highly-oxidised" (see Bianchi et al., 2019)

Line 44: If you're using Extremely Low Volatility here to refer to molecules that fall into the volatility class $C*(300K)$ $3\cdot10^{-9} < C*(300K) < 3\cdot10^{-5}$ µg m$^{-3}$, then it is worth note that by current classification of HOMs as discussed in Bianchi et al., 2019, HOMs fall across many of these volatility classes. As you refer to nucleation on line 47, I would consider revising this statement as it is currently thought that only ULVOC (as defined by Schervish & Donahue, 2019; Simon et al., 2020) are capable of undergoing pure biogenic nucleation. I would include this also in your discussion of section 3.3.3, as many of your compounds fall into this volatility class and this could make a valuable addition to this discussion.

Line 46-47: Given the nature of the paper, it may also be worth mentioning recent work discussing formation of HOMs from chlorine oxidation also (Wang et al., 2020).

Line 89: It would be nice if you provided more detail here for those unfamiliar with I-NPF. How does the evolution of the size distribution differ from conventional "banana-plot" NPF? This definition feels very arbitrary. Is there also any air-mass data that can support the assignment of "I-NPF"? It would be very beneficial to have a steadfast criteria that separates the two.

Table 1: Consider using the same units (µg m$^{-3}$) for all quantities.

Line 208-209: "The formula number of the least common CHN$^+$ group is only 46". Do you mean CHN$^+$ is the least common group?

Line 214-216: Is there any evidence for the role of amines in iodine nucleation? A reference to this would be informative here, otherwise it is of little relevance given the section discusses iodine nucleation

Line 301-303: It is not evident to me that these spectra indicate oxidation products of continental terpene emissions, the peaks at 9-10, 13-15 and 18-20 carbon numbers that would typically be expected of these products are not present. This point could do with expanding.

**Technical corrections:**

Line 44: High O:C ratio**s** (rather than "ratio")

Line 72-73: Change "Studies investigating coastal organic aerosols have been rarely" to "...aerosols are rare"

Line 80: Missing hyphen in FT-ICR-MS

Line 86: Space between 200 and m

Figure S1: I would urge the authors to consider using a colour scale that does not "cap out" at 1,500x64 cm$^{-3}$ as this disingenuously misrepresents the I-NPF event.

Line 149: Should this read "Nanometer *sized* new particles"?

Figure S2 and throughout: Dalton is not a unit of mass/charge.

Line 222: Should this read (free) troposphere?

Line 253: change to "**a** relatively high unsaturation degree"

**References**
Bianchi, F., Kurtén, T., Riva, M., Mohr, C., Rissanen, M. P., Roldin, P., Berndt, T., Crounse, J. D., Wennberg, P. O., Mentel, T. F., Wildt, J., Junninen, H., Jokinen, T., Kulmala, M., Worsnop, D. R., Thornton, J. A., Donahue, N., Kjaergaard, H. G., & Ehn, M. (2019). Highly Oxygenated Organic Molecules (HOM) from Gas-Phase Autoxidation Involving Peroxy Radicals: A Key Contributor to Atmospheric Aerosol. *Chemical Reviews*.
Mutzel, A., Poulain, L., Berndt, T., Iinuma, Y., Rodigast, M., Böge, O., Richters, S., Spindler, G., Sipila¨, M., Jokinen, T., Kulmala, M., & Herrmann, H. (2015). Highly Oxidized Multifunctional Organic Compounds

Observed in Tropospheric Particles: A Field and Laboratory Study. *Environmental Science and Technology*, *49*(13), 7754–7761. https://doi.org/10.1021/acs.est.5b00885

Peräkylä, O., Riva, M., Heikkinen, L., Quéléver, L., Roldin, P., & Ehn, M. (2019). Experimental investigation into the volatilities of highly oxygenated organic molecules (HOM). *Atmospheric Chemistry and Physics Discussions*, 1–28. https://doi.org/10.5194/acp-2019-620

Pospisilova, V., Lopez-Hilfiker, F. D., Bell, D. M., El Haddad, I., Mohr, C., Huang, W., Heikkinen, L., Xiao, M., Dommen, J., Prevot, A. S. H., Baltensperger, U., & Slowik, J. G. (2020). On the fate of oxygenated organic molecules in atmospheric aerosol particles. *Science Advances*, *6*(11), 1–12. https://doi.org/10.1126/sciadv.aax8922

Schervish, M., & Donahue, N. M. (2019). Peroxy Radical Chemistry and the Volatility Basis Set. *Atmos. Chem. Phys. Discuss*. https://doi.org/10.5194/acp-2019-509

Simon, M., Dada, L., Heinritzi, M., Scholz, W., Stolzenburg, D., Wagner, A. C., Kürten, A., Rörup, B., He, X., Almeida, J., Baccarini, A., Bauer, P. S., Beck, L., Bergen, A., Bianchi, F., Brilke, S., Caudillo, L., Chen, D., Chu, B., … Yan, C. (2020). Molecular understanding of new-particle formation from alpha-pinene between -50 °C and 25 °C. *Atmospheric Chemistry and Physics Discussions*, *January*, 1–42.

Sipilä, M., Sarnela, N., Jokinen, T., Henschel, H., Junninen, H., Kontkanen, J., Richters, S., Kangasluoma, J., Franchin, A., Peräkylä, O., Rissanen, M. P., Ehn, M., Vehkamäki, H., Kurten, T., Berndt, T., Petäjä, T., Worsnop, D., Ceburnis, D., Kerminen, V. M., … O'Dowd, C. (2016). Molecular-scale evidence of aerosol particle formation via sequential addition of HIO3. *Nature*, *537*(7621), 532–534. https://doi.org/10.1038/nature19314

Wang, Y., Riva, M., Xie, H., Heikkinen, L., Schallhart, S., Zha, Q., Yan, C., He, X., Peräkylä, O., & Ehn, M. (2020). Formation of highly oxygenated organic molecules from chlorine atom initiated oxidation of alpha-pinene. *Atmospheric Chemistry and Physics*, *20*, 5145–5155. https://doi.org/10.5194/acp-2019-807

Zhao, R., Kenseth, C. M., Huang, Y., Dalleska, N. F., Kuang, X. M., Chen, J., Paulson, S. E., & Seinfeld, J. H. (2018). Rapid Aqueous-Phase Hydrolysis of Ester Hydroperoxides Arising from Criegee Intermediates and Organic Acids. *Journal of Physical Chemistry A*, *122*(23), 5190–5201. https://doi.org/10.1021/acs.jpca.8b02195

---

## Author Response (AR1)

**Point-to-point responses to anonymous Reviewers**
by Huan Yu, yuhuan@cug.edu.cn

We appreciate two anonymous referees for their insightful comments and careful examination of the manuscript. Below, we provide point-by-point responses to their comments and questions. The manuscript is revised accordingly.

Note: all the chemical formula data used in the manuscript can be accessed in "Global Change Research Data Publishing and Repository" via DOI:10.3974/geodb.2020.03.26.V1. This statement has been added to Data Availability in the finalized manuscript.

**Reviewer comments 1**

Review of Wan et al.:
The paper presents a thorough report on the observation of a new group of aerosol organic species in coastal new iodine particles. Further effort is directed towards identifying the nature of such organic compounds, and their different contribution to size-resolved aerosol samples. The paper is very detailed and very well written, although some language editing may be beneficial. This paper fits well within the scope of ACP. I would like to congratulate the authors for this novel iodine research and I recommend publication after the comments below are addressed:

- The authors argue the source of the reported organics is likely the same as for iodine emissions (i.e. algae exposed to the atmosphere at low tide). However, they should elaborate more about this since the results shown in the paper are not clear on this regard.

For instance, do they see similar levels of OC at high tide when no iodine emissions occur? At low tide, the instrument is between 40 and 200 m from the coastal line and the emission area. Can the authors comment on the effect of this distance on their measurements? Can they see gradual differences in aerosols composition as the water recedes?

Re:
   Using our off-line techniques, only time-averaged mass concentration and chemical composition of OC were obtained for a period of three consecutive days. We thus cannot differentiate OC levels and composition between high tide and low tide. However, we have two indirect evidences to support that the OC observed in our manuscript was due to the photochemistry of low tide emissions: first, the online instruments NAIS and SMPS indicate indeed that number concentration of new particles was anti-correlated to tidal height, as show in our new Figure S2a below.

Second, during the three overcast non-NPF days from April 16 to 18, both total OC level and chemical composition in 10-18 nm size range were below or close to field blank.

As our future work, we are planning to conduct online measurement of chemical composition of clusters or nanoparticles at the site.

[Figure]

OC measured in iodine particles? Overall, the paper would benefit from some more detailed discussion on the possible sources of OC and the potential anthropogenic influence. This is important to be able to extrapolate their results to other iodine-rich coastal locations.

Re:
We add an aerial photo (Fig. S1a) to show the environment around the sampling site. Some description of the site is added in line 91-9 :

"*The sampling site (29°29' N, 121°46' E) is near a small fishing village without permanent residents in the coastline of East China Sea. It can be seen from the aerial photo (Figure S1a) that from the east to the west are the sea, intertidal zone, small paddy fields and the mountain.*"

In a new Fig S1b, we plot back trajectories during the I-NPF days from May 8 to 10, 2018.

The following discussion is added in line 156-161:

"*...strong I-NPF events were observed almost every sunny day in April and May, which was the growth and farming season of seaweed. HYSPLIT Back-trajectories analysis (Draxler and Rolph, 2010) shows that air masses moved from East China Sea to the sampling site during the I-NPF days from May 8 to 10, 2018 (Figure S1b). Sea breeze was also expected to flow from the sea to the site in the daytime when the I-NPF events occurred.*"

In line 358-363:

"*Obviously, $C_{18,30}H_hO_oN_n$ and $C_{20,24,28,33}H_hO_o$ formulas cannot be attributed to continental terpene emission or anthropogenic aromatic emissions. Sporadic spikes of*

*10-18 nm particles that can be an indication of cooking and traffic emissions were not seen in the PNSD spectrum, because such human activities were rare around the site during the sampling period. We thus also exclude the possibility of cooking and traffic emissions.*"

In summary, considering the unique OC chemical formulas (i.e. $C_{18,30}H_hO_oN_n$ and $C_{20,24,28,33}H_hO_o$) and their co-presence with iodine in the new particles, we suggest the results in our location can be extrapolated to other iodine-rich coastal locations, as long as iodine-NPF could be observed.

[Figure]

Figure S1. (a) The observation site, indicated as a red star, in an aerial photograph. Photo source: Baidu Map. (b) 72-hour air mass back trajectories ending at 100 m above ground level at the observation site computed by HYSPLIT model during the I-NPF events from May 8 to 10, 2018.

- Line 49, beginning of paragraph. It would be useful for the reader to clarify that organic iodine is not the main source of the iodine oxide precursors that lead to the formation of I-NPF. This was shown in the past for coastal regions: A. Saiz-Lopez and J. M. C. Plane, Novel iodine chemistry in the marine boundary layer. Geophys. Res. Lett. 31, L04112 (2004). and for opean ocean conditions: A. S. Mahajan et al., Measurement and modelling of tropospheric reactive halogen species over the tropical Atlantic Ocean. Atmos. Chem. Phys. 10, 4611-4624 (2010). A. S. Mahajan et al., Latitudinal distribution of reactive iodine in the Eastern Pacific and its link to open ocean sources, Atmospheric Chemistry and Physics, 12, 11609-11617 (2012).

Re:
The sentence is revised and the references are cited in line 51-57:
"*Unlike the deep understanding of continental HOMs, little is known about the role of organics in the NPF in coastal or open ocean atmosphere. The current state of knowledge is that the photolysis of molecular iodine ($I_2$) or iodomethane is the source of iodine oxides or oxoacids, the self-clustering of which could initiate NPF events with particle number concentration sometimes exceeding $10^6$ cm$^{-3}$ (O'Dowd et al., 2002; Saiz-Lopez and Plane, 2004; Burkholder et al., 2004; Mahajan et al., 2010,*

*2012; Sipilä et al., 2016; Stevanović et al., 2019; Kumar et al., 2018).*"

- Final comment. I would suggest to add a sentence to the Conclusions about the potential relevance of extrapolating the results of this location to other coastal locations, based on the questions above.

Re:
The following statement is added in the conclusion part line 543-548:
"*This article reveals a new group of important organic compounds involved in this process. It is most likely that their precursors are emitted mutually with iodine from either direct exposure of coastal biota to the atmosphere or biological-active sea surface. If this is true, we suggest the results in our location can be extrapolated to other iodine-rich coastal locations, as long as iodine-NPF can be observed.*"

**Reviewer comments 2**

**Review of Wan et al., 2020:**
This paper contains reports on the multitude of organic compounds present in size segregated aerosol samples at a coastal location where frequent new particle formation (NPF) was observed, as measured by a powerful off-line mass spectrometric technique. Iodine-NPF events are identified (I-NPF) and these are linked to aerosol chemical composition during periods with three days of consecutive I-NPF days. These are compared with similar data for C-NPF days. The potential sources are discussed, and volatilities of these compounds as calculated by a group contribution method are presented and discussed with a view to linking these compounds to new particle growth. The paper is well within the remit of ACP and contains a very valuable dataset in a field where not much is yet known, however, I find issues with the authors' definition of I-NPF, and the discussion of calculated vapour pressures could do with some extension as there are many uncertainties with these calculations not discussed. I recommend publication after the following issues are addressed.

**General comments:**
- In **section 3.1** the authors present 10-56 nm particle phase composition in 72 hour aerosol samples through 3 consecutive NPF days of two different categories (iodine-induced NPF, or *I-NPF*, & continental regional NPF, or *C-NPF*), identified from size-distribution measurements. It is evident from the size distributions that these NPF days *are* very dissimilar, and the evolution of the size distribution is markedly similar to prior reports of iodine-nucleation (such as Sipilä et al., 2016), however, the authors state that an elevation to the mass concentration of iodine in 10-56 nm aerosol samples during the I-NPF days is clear indication that NPF was linked to iodine nucleation. I would strongly argue this is not a clear indication a nucleation process involving iodine vapours, as aerosol mass between 10-56 nm is not an indicator of what process produces particles at ~1.7 nm. I would urge the authors to back this claim up with reference to similar reports of iodine driven nucleation under conditions that produce either similar particle composition or similar size distributions. $HSO_4^-$ concentrations are still an order of magnitude higher than $I^-$ concentrations in 10-18 nm particles on I-NPF days. If there is there any evidence within the data that nucleation processes are not dominated, for example, by sulphuric acid processes, this would strengthen this section greatly.

Re:
In the revised manuscript we list four facts following the suggestions of the reviewer. In line 154-173, the revised part reads as follows:

*"Although our offline technique did not allow us to probe nucleating cluster composition at ~1.7 nm, four facts from our observation support that the NPF events from May 9 to 11 were initiated by iodine nucleation. First, strong I-NPF events were*

*observed almost every sunny day in April and May, which was the growth and farming season of seaweed. HYSPLIT Back-trajectories analysis (Draxler and Rolph, 2010) shows that air masses moved from East China Sea to the sampling site during the I-NPF days from May 8 to 10, 2018 (Figure S1b). Sea breeze was also expected to flow from the sea to the site in the daytime when the I-NPF events occurred. Second, the evolution of PNSD from May 9 to 11 was not like banana-shape C-NPF observed on the winter days, but was markedly similar to prior reports of iodine-nucleation at European coastal sites (Mäkelä et al., 2002; Sipilä et al., 2016). Third, the production of 2-7 nm particles ($N_{2-7}$) during the C-NPFs followed a nearly identical variation with solar radiation (Figure S2c), which is an indication that the C-NPFs was initiated by OH and $H_2SO_4$ production dictated by solar radiation. However, this was not observed during the I-NPF events, instead, $N_{2-7}$ was anti-correlated to tidal height in the daytime (Figure S2a). Fourth, probably the most important, mean total I in 10-56 nm particles during the I-NPF days (13.5 ng $m^{-3}$, Table 1) was 67 and 36 times higher than those during the C-NPF days (0.2 ng $m^{-3}$) and non-event days (0.37 ng $m^{-3}$). In the same size range, mean $HSO_4^-$ concentration (0.2 μg $m^{-3}$) during the I-NPF days was lower than that during the C-NPF days (0.5 μg $m^{-3}$)."*

Another comparison with previous iodine studies in terms of chemical composition is added in line 182-185:

*...Our result is qualitatively consistent with previous measurements showing that nucleation mode particles initiated by iodine were composed of a remarkable fraction of organics and sulfate (Mäkelä et al., 2002; Vaattovaara et al., 2006).*"

The time series of $N_{2-7}$, tidal height and solar radiation are added to Figure S2.

[Figure]

Figure S2. (a), (c) Number concentration of 2-7 nm particles ($N_{2-7}$), tidal height and solar radiation intensity during the Iodine-initiated NPF (I-NPF) days from May 9 to 11 and the continental regional NPF (C-NPF) days from February 11 to 13. Particle number size distribution during (b) I-NPF days from May 9 to 11, (d) C-NPF days from February 11 to 13 and (e) non-NPF days from April 16 to 18.

- It would be very useful to mention the magnitude of the effect of these NPF events to aerosol mass in the relevant size ranges – coming up with an exact number is a rather uncertain process, but a simple alternative would be a plot showing the time evolution of size-segregated aerosol mass as currently there is no indication of how much of the aerosol mass is actually arising from NPF.

Re:
The time series of mass concentration of 10-56 nm particles was estimated from the PNSD data by assuming a particle density of 1.5 g cm$^{-3}$. The time series was added to Figure S2.

The inserted sentences in line 174-182 read as

*"By assuming nanometer sized particles are spherical with a density of 1.5 g cm$^{-3}$, we estimate from the PNSD data that aerosol mass in the 10-56 nm size range was enhanced by 3.0 and 1.3 μg m$^{-3}$ at most by the selected I-NPF and C-NPF events (Figure S2b and S2d). The fraction of organic mass (OM) in the aerosol mass can be further calculated from $(1.5 \times m_{TOC})/(m_{Total\ I} + m_{HSO4^-} + 1.5 \times m_{TOC}) \times 100\%$ by assuming an OM/TOC ratio of 1.5. The result shows that mass fractions of OM are 95%, 87% and 68%, respectively, in the size bins 10-18 nm, 18-32 nm and 32-56 nm during the I-NPF days. Therefore, organic compounds dominate the aerosol mass in the 10-56 nm new particles during the I-NPF days and were critical for I-NPF to contribute to CCN."*

- Recent chamber results utilising EESI-ToF-MS would indicate that oligermisation processes in the particle phase can produce a diverse range of compounds (Pospisilova et al., 2020), as well as particle phase processes producing organosulphates (Mutzel et al., 2015), formation of ester hydroperoxides (Zhao et al., 2018) etc. I would consider how such mechanisms may affect your proposed formation mechanisms, and further your estimations of volatility - if some of these compounds arise from particle-phase oligomerisation, does this change the conclusions of the paper? I would like to echo reviewer #1 here and suggest an extension of the discussion so the results can be more easily extrapolated to other coastal regions.

Re:
    According to the paper by Pospisilova et al., 2020 and others, both monomers (C5–C10) and dimers (C15–C20) are evident in chamber organic aerosols produced by alpha-pinene ozonolysis. Moreover, the mass spectrum is dominated by monomer signals. We didn't see such monomer-dimer distribution pattern in our mass spectrum for either $C_{18,30}H_hO_oN_n$ or $C_{20,24,28,33}H_hO_o$ formulas.

    Multifunctional organic peroxides, such as αAAHPs, may comprise an important but unresolved fraction of SOA (Zhao et al. 2018 a,b). These studies also demonstrated that organic peroxides may undergo fast hydrolysis or decomposition to end products containing carboxyl, carbonyl or hydroxyl groups. A basic assumption in our study is the formulas we observed are end products. Our volatility estimation thus considered only end products containing $-ONO_2$, $-OH$, $-C=O$, $-NH_2$ and $-COOH$ groups. The formation of those intermediates like αAAHPs in the gas phase and air-water interface is possible, but not measurable by our methods.

However, the missing information about possible intermediates will not change the conclusion of our manuscript, that is, the addition of $-ONO_2$, $-OH$, $-C=O$, and $-COOH$ groups to biogenic precursors reduced the volatility of precursors by 2~7 orders of magnitude and the end products containing $-ONO_2$, $-OH$, $-C=O$, $-NH_2$, $-COOH$ groups fall into the range of ELVOC and ULVOC. In fact, the formation of low volatility intermediates like αAAHPs may further enhance the partitioning of organic products to particle phase.

Organosulfate accounted for only ~10% of ESI- signals in 10-18 nm particles. Their possible formation in the gas or particle phase is beyond the focus of our manuscript. We didn't discuss their formation mechanism and volatility in the manuscript.

We add the following statement in line 325-329:

*"Monomer-dimer distribution pattern that can arise from particle-phase oligomerisation (Pospisilova et al., 2020) was not observed for these formulas in the mass spectra. We also assume that $C_{18,30}H_hO_oN_n$ and $C_{20,24,28,33}H_hO_o$ are not labile intermediates like ester hydroperoxides that may undergo fast decomposition in the particles or during the sample preparation process (Zhao et al. 2018 a,b)."*

- The estimation of volatilities through SIMPOL has some associated uncertainties, and, at least in the case of HOMs, estimations of volatility can vary by several orders of magnitude when compared with other methods (see Figure 8 of Peräkylä et al., 2019). This should be a point of discussion as it may affect the outcomes of the paper.

Re:
The volatility of VOC oxidation products can be assessed with numerous existing parameterizations, which require either exact functional groups making up a molecule or only the molecular formula. Their estimation can be consistent with each other quite well or vary by up to several orders of magnitude. At this moment, we cannot declare which is better than others. For example, the equation given by the Peräkylä et al., 2019 method may only hold for SVOC and LVOC range, because the gas-particle partitioning behavior of the compounds outside the SVOC and LVOC range cannot be measured easily by the Peräkylä et al., 2019 method.

But this will not change the conclusion drawn in the manuscript, that is, the addition of –ONO$_2$, –OH, –C=O , and -COOH groups reduces the volatility of precursors greatly and thus make their oxidation products condensable onto new particles during the I-NPF event days.

In line 485-489 we add

"*It should be noted that the volatility of VOC oxidation products can be assessed with numerous existing parameterizations, which require either exact functional groups or only the molecular formula (Peräkylä et al., 2019). Their estimation can vary by up to several orders of magnitude. But this will not change the conclusion drawn here.*"

**Specific comments:**

Line 22 and throughout: Change "organics" to "organic compounds".

Re: corrected.

Line 43: "Highly oxygenated (multifunctional) organic molecules" is a more correct term for these than "highly-oxidised" (see Bianchi et al., 2019)

Re: corrected.

Line 44: If you're using Extremely Low Volatility here to refer to molecules that fall into the volatility class C*(300K) 3·10–9 < C*(300K) < 3·10–5 μg m–3, then it is worth note that by current classification of HOMs as discussed in Bianchi et al., 2019, HOMs fall across many of these volatility classes. As you refer to nucleation on line 47, I would consider revising this statement as it is currently thought that only ULVOC (as defined by Schervish & Donahue, 2019; Simon et al., 2020) are capable of undergoing pure biogenic nucleation. I would include this also in your discussion of section 3.3.3, as many of your compounds fall into this volatility class and this could make a valuable addition to this discussion.

Re:

Now we delete "extreme" from line 45. What we want to say here is HOMs with low volatility contributed to both nucleation/growth and SOA formation. The statement reads as follows:

"*..Recent laboratory and field studies have identified a group of highly oxygenated multifunctional organic molecules (HOMs) with high O/C ratios and low volatility… These HOMs play an important role in particle nucleation and growth of continental NPF, as well as in the formation of secondary organic aerosols.*"

In section 3.3.3 we revise the following statements:

"*As we can see in Table S4, C* of the 49 formulas fall into the range of ELVOC (3 × 10$^{-9}$ - 3 × 10$^{-5}$ μg m$^{-3}$) and even ULVOC (ultra-low volatility organic compound, <3 × 10$^{-9}$ μg m$^{-3}$), while C* of their precursors are in the range of SVOC (0.3-300 μg m$^{-3}$)*

*or LVOC (3 × 10-5-0.3 μg m-3). The addition of functional groups reduces the volatility of precursors by 2~7 orders of magnitude and thus make their oxidation products condensable onto new particles during the I-NPF event days. According to the definition of Schervish and Donahue, 2019 and Simon et al., 2020, ULVOC can even drive pure biogenic nucleation. Therefore, the analysis of precursor-product volatility partly supports our hypothesis about the molecular identity and formation mechanism of the formulas detected in 10-18 nm particles.*"

Line 46-47: Given the nature of the paper, it may also be worth mentioning recent work discussing formation of HOMs from chlorine oxidation also (Wang et al., 2020).

Re:
We add chlorine atom and cite the paper of Wang et al., 2020 in line 48.

Line 89: It would be nice if you provided more detail here for those unfamiliar with I-NPF. How does the evolution of the size distribution differ from conventional "banana-plot" NPF? This definition feels very arbitrary. Is there also any air-mass data that can support the assignment of "I-NPF"? It would be very beneficial to have a steadfast criteria that separates the two.

Re:
Please see our response to your earlier comment. We present four facts, including air mass back trajectory analysis, to support the definition of I-NPF in line 154-173, Section 3.1

Table 1: Consider using the same units (μg m-3) for all quantities.
Re:
Change ng m-3 to μg m-3

Line 208-209: "The formula number of the least common CHN+ group is only 46". Do you mean CHN+ is the least common group?
Re:
Yes, CHN+ is the least common group.

Line 214-216: Is there any evidence for the role of amines in iodine nucleation? A reference to this would be informative here, otherwise it is of little relevance given the section discusses iodine nucleation
Re:
There is no evidence or reference for the role of amines in iodine nucleation. However, generally speaking, basic amines are expected to stabilize $H_2SO_4$-$H_2O$ cluster in atmospheric NPF. People may be curious if amines can be present in our particles in such an location where marine organism can be a significant source of amine emission. So we just want to report here that amines contributed a negligible fraction to new particles in our case.

Line 301-303: It is not evident to me that these spectra indicate oxidation products of continental terpene emissions, the peaks at 9-10, 13-15 and 18-20 carbon numbers that would typically be expected of these products are not present. This point could do with expanding.

Re:

It can be clearly seen Figure 3d and Figure 2c that $CHO^-$ formulas in the 180-560 nm particles are trimodal with maximum intensity around $C_9$, $C_{13}$-$C_{16}$ and $C_{20}$, while $CHON^-$ are bimodal around $C_{10}$ and $C_{15}$.

**Technical corrections:**

Line 44: High O:C ratio**s** (rather than "ratio")

Line 72-73: Change "Studies investigating coastal organic aerosols have been rarely" to "...aerosols are rare"

Line 80: Missing hyphen in FT-ICR-MS

Line 86: Space between 200 and m

Figure S1: I would urge the authors to consider using a colour scale that does not "cap out" at 1,500x64 $cm_{-3}$ as this disingenuously misrepresents the I-NPF event.

Line 149: Should this read "Nanometer *sized* new particles"?

Figure S2 and throughout: Dalton is not a unit of mass/charge.

Line 222: Should this read (free) troposphere?

Line 253: change to "**a** relatively high unsaturation degree"

Re:

We appreciate the reviewer for his/her care examination of the manuscript. All problems are now solved following the comments above.

[revised manuscript text omitted]

**Text:**

ESI-FT-ICR MS operation conditions

**Figures:**

Figure S1: Figure S1. (a) The observation site, indicated as a red star, in an aerial photograph. Photo source: Baidu Map. (b) 72-hour air mass back trajectories ending at 100 m above ground level at the observation site computed by HYSPLIT model during the I-NPF events from May 8 to 10, 2018.

Figure S2. (a), (c) Number concentration of 2-7 nm particles ($N_{2-7}$), tidal height and solar radiation intensity during the Iodine-initiated NPF (I-NPF) days from May 9 to 11 and the continental regional NPF (C-NPF) days from February 11 to 13. Particle number size distribution and 10-56 nm particle mass concentration during (b) I-NPF days from May 9 to 11, (d) C-NPF days from February 11 to 13 and (e) non-NPF days from April 16 to 18.

Figure S3. Reconstructed mass spectra of the 7 elemental groups in ESI-and ESI+ modes for the four size bins.

Figure S4: DBE vs. C atom number diagrams of all the CHON and CHO formulas detected in 10–18 nm particles in ESI+ mode. (a) (b) +H adducts, (c) (d) +Na adducts. The color bar denotes the O number in the formulas. The size of the circles reflects the relative intensities of molecular formulas on a logarithmic scale.

Figure S5: Relative intensities of subgroups according to O atom number in CHON, CHO, CHONI and CHOI formulas in the four size bins in ESI+ (in red) and ESI- (in blue). The intensity of the most abundant subgroup is defined as 1 and those of other subgroups are normalized by it. The relative intensities of non-iodinated OC formulas (iodinated OC formulas) are plotted in the region above (below) zero line.

Figure S6: O atom number of *vs.* N atom number of $C_{10}H_xO_yN_z$ compounds detected in 180–560 nm particles (a) and $C_{18}H_xO_yN_z$ compounds detected in 10–18 nm particles in ESI- mode (b).

Figure S7: Simplified reaction scheme showing the formation of oxygenated and nitrated CHO and CHON compounds from α-linolenic acid oxidation in the atmosphere.

Figure S8: Simplified reaction scheme showing the formation of oxygenated CHO compounds from unsaturated $C_{28}$ FA ($C_{28}H_{52}O_2$) oxidation in the atmosphere.

Figure S9: Simplified reaction scheme showing the formation of oxygenated CHON compounds containing a $-NH_2$ group from unsaturated $C_{18}$ amino alcohol ($C_{18}H_{37}NO_4$) oxidation in the atmosphere.

**Tables:**

Table S1. Predicted saturation concentration (C*) range of most abundant CHON and CHO formulas, as well as their possible precursors.

**ESI-FT-ICR MS operation conditions**

A syringe pump infused the sample extract continuously into the ESI unit with a flow rate of 180 μL h-1. The ESI source conditions were as follows: the nebulizer gas pressure was 1 bar; the dry gas ($N_2$) pressure was 4 bar and its temperature was 200 ℃; the capillary voltage was 4.5 kV. The ion accumulation time in the argon-filled hexapole collision pool with 1.5 V of direct current voltage and 1400 Vp-p of radio frequency (RF) amplitude was 0.05 s, followed by transport ions through a hexapole ion guide to the ICR cell for 0.7 ms. 4 M words of data were recorded over the mass range of 150-1000 Da for each run. A total of 128 scans were collected to enhance the signal/noise (S/N) ratio and dynamic range.

[Figure]

[Figure]

Figure S1. (a) The observation site, indicated as a red star, in an aerial photograph.

Photo source: Baidu Map. (b) 72-hour air mass back trajectories ending at 100

m above ground level at the observation site computed by HYSPLIT model during the I-NPF events from May 8 to 10, 2018.

[Figure]

**a**

$N_{2-7}$ (cm$^{-3}$) · Solar radiation (W m$^{-2}$) · Tidal height (m)

**b**

$D_p$ (nm) · 10-56 nm mass conc. (μg m$^{-3}$)

0:00 5/8 · 12:00 · 0:00 5/9 · 12:00 · 0:00 5/10 · 12:00 · 0:00 5/11

**c**

$N_{2-7}$ (cm$^{-3}$) · Solar radiation (W m$^{-2}$) · Tidal height (m)

**d**

$D_p$ (nm) · 10-56 nm mass conc. (μg m$^{-3}$)

0:00 2/11 · 12:00 · 0:00 2/12 · 12:00 · 0:00 2/13 · 12:00 · 0:00 2/14

**e**

$D_p$ (nm)

0:00 4/16 · 12:00 · 0:00 4/17 · 12:00 · 0:00 4/18 · 12:00 · 0:00 4/19

**Local Time**

· 20 · 40 · 60 · 80 · 100x10$^3$

dN/dlogD$_p$ ( cm$^{-3}$)

Figure S2. (a), (c) Number concentration of 2-7 nm particles ($N_{2-7}$), tidal height and solar radiation intensity during the Iodine-initiated NPF (I-NPF) days from May 9 to 11 and the continental regional NPF (C-NPF) days from February 11 to 13. Particle number size distribution and 10-56 nm particle mass concentration during (b) I-NPF days from May 9 to 11, (d) C-NPF days from February 11 to 13 and (e) non-NPF days from April 16 to 18.

[Figure]

Figure S3. Reconstructed mass spectra of the 7 elemental groups in ESI- (left panels) and ESI+ (right panels) modes for the four size bins. The signals are normalized against the intensity of the most abundant molecular ions in a size bin.

[Figure]

Figure S4. DBE vs. C atom number diagrams of all CHON and CHO formulas detected in 10–18 nm particles in ESI+ mode. (a) (b) [M+H]$^+$ adducts, (c) (d) [M+Na]$^+$ adducts. The color bar denotes O number in the formulas. The size of the circles reflects the relative intensities of molecular formulas on a logarithmic scale.

[Figure]

Figure S5. Relative intensities of subgroups according to O atom number in CHON, CHO, CHONI and CHOI formulas in the four size bins in ESI+ (in red) and ESI- (in blue). The intensity of the most abundant subgroup in a size bin is defined as 1 and those of other subgroups are normalized by it. The relative intensities of non-iodinated OC formulas (iodinated OC formulas) are plotted in the region above (below) zero line.

[Figure]

Figure S6. O atom number of *vs.* N atom number of $C_{10}H_hO_oN_n^-$ compounds detected in 180–560 nm particles (a) and $C_{18}H_hO_oN_n$- compounds detected in 10–18 nm particles in ESI- mode (b).

[Figure]

Figure S7. Simplified reaction scheme of the formation of oxygenated and nitrated CHO and CHON compounds from α-linolenic acid ($C_{18}H_{30}O_2$) oxidation in the atmosphere. One representative structure is shown for each chemical formula. Chemical formulas in the boxes are found in the formula list detected in 10–18 nm particles. Pathway 1: OH and $O_2$ addition followed by reaction with NO to form a – $ONO_2$ group; pathway 2: OH and $O_2$ addition followed by reaction with NO to form an alkoxy radical that further reacts with $O_2$ to form a –C=O group.

[Figure]

Figure S8. Simplified reaction scheme of the formation of oxygenated CHO compounds from unsaturated $C_{28}$ FA ($C_{28}H_{52}O_2$) oxidation in the atmosphere. One representative structure is shown for each chemical formula. Chemical formulas in the boxes are found in the formula list detected in 10–18 nm particles. Pathway 3: OH and $O_2$ addition followed by reaction with $RO_2$ to form a –OH or a –C=O group; Pathway 4: successive intermolecular H-shift/$O_2$ addition (autoxidation) to form $RO_2$ radicals with –OOH group. –OOH group is not stable and decomposed to -OH.

[Figure]

Figure S9. Simplified reaction scheme of the formation of oxygenated CHON compounds containing a –NH$_2$ group from unsaturated C$_{18}$ amino alcohol (C$_{18}$H$_{37}$NO$_4$) oxidation in the atmosphere. One representative structure is shown for each chemical formula. Chemical formulas in the boxes are found in the formula list detected in 10–18 nm particles. Pathway 3: OH and O$_2$ addition followed by reaction with RO$_2$ to form a –OH

or a –C=O group; Pathway 4: successive intermolecular H-shift/O$_2$ addition autoxidation to form RO$_2$ radicals with –OOH group.

Table S1. Predicted saturation concentration (C*) range of most abundant CHON and CHO formulas, as well as their possible precursors.

| Formula | Predicted C* ($\mu g \ m^{-3}$) | Predicted C* of possible precursors ($\mu g \ m^{-3}$) |
|---|---|---|
| ESI- mode | | |
| $C_{18}H_{33}NO_4$ | $1.62 \times 10^{-5}$–$2.06 \times 10^{-2}$ | $3.40 \times 10^{-1}$–$8.91$ |
| $C_{18}H_{33}NO_6$ | $7.66 \times 10^{-10}$–$1.33 \times 10^{-2}$ | $3.40 \times 10^{-1}$–$8.87 \times 10^1$ |
| $C_{18}H_{34}N_2O_6$ | $7.62 \times 10^{-11}$–$1.32 \times 10^{-3}$ | $3.40 \times 10^{-2}$–$8.91$ |
| $C_{18}H_{34}N_2O_7$ | $5.21 \times 10^{-13}$–$9.06 \times 10^{-6}$ | $3.40 \times 10^{-2}$–$8.91$ |
| $C_{18}H_{34}N_2O_8$ | $1.30 \times 10^{-15}$–$5.56 \times 10^{-6}$ | $3.40 \times 10^{-2}$–$8.91$ |
| $C_{18}H_{36}N_2O_5$ | $1.44 \times 10^{-8}$–$1.10 \times 10^{-2}$ | $3.40 \times 10^{-2}$–$8.87 \times 10^1$ |
| $C_{18}H_{36}N_2O_6$ | $9.83 \times 10^{-11}$–$7.54 \times 10^{-5}$ | $3.40 \times 10^{-2}$–$8.87 \times 10^1$ |
| $C_{18}H_{36}N_2O_7$ | $6.72 \times 10^{-13}$–$5.15 \times 10^{-7}$ | $3.40 \times 10^{-2}$–$8.87 \times 10^1$ |
| $C_{19}H_{39}NO_7$ | $3.40 \times 10^{-12}$–$1.15 \times 10^{-7}$ | $1.34 \times 10^{-1}$–$3.51 \times 10^1$ |
| $C_{30}H_{57}NO_4$ | $7.44 \times 10^{-8}$–$6.69 \times 10^{-6}$ | $4.29 \times 10^{-6}$–$1.16 \times 10^{-3}$ |
| $C_{30}H_{59}NO_3$ | $5.58 \times 10^{-9}$–$1.58 \times 10^{-6}$ | $4.29 \times 10^{-6}$–$1.14 \times 10^{-4}$ |
| $C_{30}H_{59}NO_4$ | $6.28 \times 10^{-10}$–$8.62 \times 10^{-6}$ | $4.29 \times 10^{-6}$–$1.16 \times 10^{-3}$ |
| $C_{30}H_{59}NO_5$ | $4.52 \times 10^{-12}$–$1.32 \times 10^{-6}$ | $4.29 \times 10^{-6}$–$1.16 \times 10^{-3}$ |
| $C_{30}H_{59}NO_6$ | $2.64 \times 10^{-13}$–$8.93 \times 10^{-9}$ | $4.29 \times 10^{-6}$–$1.16 \times 10^{-3}$ |
| $C_{30}H_{60}O_6$ | $1.66 \times 10^{-14}$–$3.76 \times 10^{-13}$ | $4.36 \times 10^{-5}$–$1.16 \times 10^{-3}$ |
| $C_{20}H_{40}O_6$ | $2.13 \times 10^{-10}$–$4.83 \times 10^{-9}$ | $5.28 \times 10^{-1}$–$1.39 \times 10^1$ |
| $C_{21}H_{42}O_6$ | $8.32 \times 10^{-11}$–$1.88 \times 10^{-9}$ | $2.07 \times 10^{-1}$–$5.46 \times 10^0$ |
| $C_{22}H_{44}O_4$ | $6.95 \times 10^{-7}$–$1.57 \times 10^{-5}$ | $8.15 \times 10^{-2}$–$2.15 \times 10^0$ |
| $C_{24}H_{48}O_4$ | $1.06 \times 10^{-7}$–$2.39 \times 10^{-6}$ | $1.25 \times 10^{-2}$–$3.3 \times 10^{-1}$ |
| $C_{26}H_{52}O_4$ | $1.60 \times 10^{-8}$–$3.63 \times 10^{-7}$ | $1.90 \times 10^{-3}$–$5.05 \times 10^{-2}$ |
| $C_{27}H_{54}O_6$ | $2.87 \times 10^{-13}$–$6.49 \times 10^{-12}$ | $7.42 \times 10^{-4}$–$1.97 \times 10^{-2}$ |
| $C_{28}H_{56}O_4$ | $2.41 \times 10^{-9}$–$5.47 \times 10^{-8}$ | $2.89 \times 10^{-4}$–$7.67 \times 10^{-3}$ |
| $C_{28}H_{56}O_6$ | $1.11 \times 10^{-13}$–$2.51 \times 10^{-12}$ | $2.89 \times 10^{-4}$–$7.67 \times 10^{-3}$ |
| $C_{29}H_{58}O_6$ | $4.29 \times 10^{-14}$–$9.73 \times 10^{-13}$ | $1.12 \times 10^{-4}$–$2.98 \times 10^{-3}$ |
| $C_{33}H_{66}O_6$ | $9.56 \times 10^{-16}$–$2.17 \times 10^{-14}$ | $2.54 \times 10^{-6}$–$6.77 \times 10^{-5}$ |
| $C_{38}H_{76}O_8$ | $3.66 \times 10^{-22}$–$8.30 \times 10^{-21}$ | $2.18 \times 10^{-8}$–$5.85 \times 10^{-7}$ |
| ESI+ mode | | |
| $C_{11}H_{18}N_4O_8$ | $4.73 \times 10^{-9}$–$3.63 \times 10^{-3}$ | $2.21 \times 10^0$–$5.61 \times 10^2$ |
| $C_{12}H_{20}N_4O_8$ | $1.85 \times 10^{-9}$–$1.42 \times 10^{-3}$ | $8.85 \times 10^{-1}$–$2.26 \times 10^2$ |
| $C_{19}H_{35}NO_3$ | $9.26 \times 10^{-4}$ | $1.34 \times 10^{-1}$–$2.23 \times 10^{-1}$ |
| $C_{19}H_{36}N_2O_5$ | $4.35 \times 10^{-9}$–$3.34 \times 10^{-3}$ | $1.34 \times 10^{-1}$–$3.52 \times 10^0$ |
| $C_{19}H_{37}NO_3$ | $1.20 \times 10^{-3}$–$2.71 \times 10^{-2}$ | $1.34 \times 10^{-1}$–$2.23 \times 10^{-1}$ |

| | | |
|---|---|---|
| $C_{19}H_{38}N_2O_3$ | $2.71 \times 10^{-3}$–$5.04 \times 10^{-3}$ | $1.34 \times 10^{-2}$–$3.52 \times 10^{0}$ |
| $C_{24}H_{46}N_2O_4$ | $5.73 \times 10^{-9}$–$4.39 \times 10^{-3}$ | $1.24 \times 10^{-4}$–$3.28 \times 10^{-2}$ |
| $C_{25}H_{43}NO_4$ | $3.07 \times 10^{-7}$–$1.72 \times 10^{-5}$ | $4.83 \times 10^{-4}$–$1.28 \times 10^{-2}$ |
| $C_{26}H_{51}NO_5$ | $1.73 \times 10^{-9}$ | $1.88 \times 10^{-4}$–$3.13 \times 10^{-4}$ |
| $C_{27}H_{50}N_2O_4$ | $6.42 \times 10^{-10}$–$3.29 \times 10^{-7}$ | $7.23 \times 10^{-6}$–$1.95 \times 10^{-3}$ |
| $C_{27}H_{50}N_2O_5$ | $3.11 \times 10^{-11}$–$1.05 \times 10^{-6}$ | $7.23 \times 10^{-6}$–$1.95 \times 10^{-3}$ |
| $C_{27}H_{52}N_2O_3$ | $4.94 \times 10^{-8}$–$2.76 \times 10^{-6}$ | $7.23 \times 10^{-6}$–$1.95 \times 10^{-3}$ |
| $C_{28}H_{52}N_2O_6$ | $1.14 \times 10^{-14}$–$8.03 \times 10^{-8}$ | $2.81 \times 10^{-6}$–$7.57 \times 10^{-4}$ |
| $C_{28}H_{54}N_2O_6$ | $5.95 \times 10^{-15}$–$1.03 \times 10^{-7}$ | $2.81 \times 10^{-6}$–$7.57 \times 10^{-4}$ |
| $C_{28}H_{56}N_2O_3$ | $6.09 \times 10^{-8}$–$1.38 \times 10^{-6}$ | $2.81 \times 10^{-6}$–$7.57 \times 10^{-4}$ |
| $C_{28}H_{56}N_2O_6$ | $1.89 \times 10^{-14}$–$5.88 \times 10^{-9}$ | $2.81 \times 10^{-6}$–$7.57 \times 10^{-4}$ |
| $C_{28}H_{58}N_2O_3$ | $3.18 \times 10^{-8}$ | $2.81 \times 10^{-6}$–$4.66 \times 10^{-6}$ |
| $C_{29}H_{56}N_2O_6$ | $2.30 \times 10^{-15}$–$3.99 \times 10^{-8}$ | $1.09 \times 10^{-6}$–$2.94 \times 10^{-4}$ |
| $C_{29}H_{59}NO_7$ | $2.85 \times 10^{-17}$–$6.47 \times 10^{-16}$ | $1.11 \times 10^{-5}$–$2.94 \times 10^{-4}$ |
| $C_{33}H_{59}NO_5$ | $1.05 \times 10^{-12}$–$3.02 \times 10^{-8}$ | $2.49 \times 10^{-7}$–$6.65 \times 10^{-6}$ |
| $C_{34}H_{59}NO_6$ | $2.13 \times 10^{-15}$–$1.38 \times 10^{-9}$ | $9.64 \times 10^{-8}$–$2.57 \times 10^{-6}$ |
| $C_{34}H_{66}N_2O_3$ | $1.15 \times 10^{-11}$–$3.58 \times 10^{-9}$ | $9.45 \times 10^{-9}$–$2.57 \times 10^{-6}$ |
| $C_{34}H_{68}N_2O_3$ | $2.03 \times 10^{-10}$–$4.61 \times 10^{-9}$ | $9.45 \times 10^{-9}$–$2.57 \times 10^{-6}$ |
| $C_{34}H_{68}N_2O_5$ | $3.75 \times 10^{-15}$–$2.88 \times 10^{-9}$ | $9.45 \times 10^{-9}$–$2.57 \times 10^{-6}$ |